# Collagen promotes anti-PD-1/PD-L1 resistance in cancer through LAIR1-dependent CD8+ T cell exhaustion

David H. Peng[1], Bertha Leticia Rodriguez[1], Lixia Diao [2], Limo Chen[1], Jing Wang [2], Lauren A. Byers [1], Ying Wei[3], Harold A. Chapman [3], Mitsuo Yamauchi[4], Carmen Behrens[5], Gabriela Raso [5], Luisa Maren Solis Soto [5], Edwin Roger Parra Cuentes [5], Ignacio I. Wistuba[5], Jonathan M. Kurie[1] & Don L. Gibbons[1,6 ✉]

Tumor extracellular matrix has been associated with drug resistance and immune suppression. Here, proteomic and RNA profiling reveal increased collagen levels in lung tumors resistant to PD-1/PD-L1 blockade. Additionally, elevated collagen correlates with decreased total CD8+ T cells and increased exhausted CD8+ T cell subpopulations in murine and human lung tumors. Collagen-induced T cell exhaustion occurs through the receptor LAIR1, which is upregulated following CD18 interaction with collagen, and induces T cell exhaustion through SHP-1. Reduction in tumor collagen deposition through LOXL2 suppression increases T cell infiltration, diminishes exhausted T cells, and abrogates resistance to anti-PD-L1. Abrogating LAIR1 immunosuppression through LAIR2 overexpression or SHP-1 inhibition sensitizes resistant lung tumors to anti-PD-1. Clinically, increased collagen, LAIR1, and TIM-3 expression in melanoma patients treated with PD-1 blockade predict poorer survival and response. Our study identifies collagen and LAIR1 as potential markers for immunotherapy resistance and validates multiple promising therapeutic combinations.

[1] Department of Thoracic/Head and Neck Medical Oncology, The University of Texas MD Anderson Cancer Center, Houston, TX 77030, USA. [2] Department of Bioinformatics and Computational Biology, The University of Texas MD Anderson Cancer Center, Houston, TX 77030, USA. [3] Department of Medicine, UCSF Cardiovascular Research Institute, San Francisco, CA, USA. [4] Oral and Craniofacial Health Sciences, University of North Carolina at Chapel Hill, Chapel Hill, NC, USA. [5] Department of Translational Molecular Pathology, The University of Texas MD Anderson Cancer Center, Houston, TX 77030, USA. [6] Department of Molecular and Cellular Oncology, The University of Texas MD Anderson Cancer Center, Houston, TX 77030, USA. ✉email: dlgibbon@mdanderson.org

Lung cancer is the leading cause of cancer-associated deaths worldwide due to late-stage disease presentation, metastasis, and resistance to therapies[1,2]. A defining characteristic of cancer is the ability to evade immune destruction through multiple mechanisms, including dysregulation of immune checkpoint pathways[3]. Thus, inhibition of immune checkpoint molecules through PD-1 or PD-L1 blockade has demonstrated significant clinical advancements for lung cancer patients[4–7]. Despite improvements in overall patient survival, immunotherapies fail to produce durable response in a majority of patients, suggesting innate and adaptive resistance mechanisms. Although several non-redundant tumor and immune cell-intrinsic mechanisms of immunotherapy resistance have been described, the role of the extracellular matrix (ECM) in immune checkpoint blockade resistance is poorly defined[8–11]. Therefore, elucidating the comprehensive mechanisms of resistance to PD-1/PD-L1 axis blockade will identify potential biomarkers to predict patient response and validate promising combinatorial therapies to improve patient survival.

Previous work by our group utilizing immune-competent syngeneic lung tumor models derived from $Kras^{LA1-G12D};$ $p53^{R172H}$ (KP) mutant mice demonstrated that KP lung cancer cells have elevated levels of PD-L1[12], consistent with analyses from lung cancer patient datasets[13]. However, PD-(L)1 blockade in KP GEM mice showed only transient effects, without a long-term reduction in primary lung tumor growth or improvement in animal survival[8]. In addition to high PD-L1 expression, our prior work also demonstrated that KP lung tumors have increased LOXL2 crosslinking, which stabilizes and enhances the deposition of collagen, a main component of the ECM that has been implicated in promoting lung tumor progression, metastasis and drug resistance[14–17]. Furthermore, studies have also correlated TGF-β signaling and TGF-β-associated ECM gene signatures, such as collagen, with tumor immune suppression and anti-PD-1/PD-L1 resistance[18,19]. Despite these observations, TGF-β is a pleiotropic molecule with multiple downstream functions and acts as a tumor suppressor or promoter depending on the context[20–22]. In addition, the precise mechanism of immune suppression and anti-PD-1/PD-L1 resistance by tumor-associated collagen has not been comprehensively investigated.

Here, we demonstrate that lung tumors which possess inherent or acquired resistance to PD-1/PD-L1 blockade have higher collagen deposition, resulting in tumor immune suppression characterized by decreased total intratumoral CD8$^+$ T cells—the lymphocytes primarily responsible for immune-mediated tumor cell death[8,12,23]—and increased TIM-3$^+$ exhausted CD8$^+$ T cell subpopulations in murine and human lung tumors. Mechanistically, collagen-induced CD8$^+$ T cell exhaustion is due to the leukocyte-specific collagen receptor LAIR1, which suppresses lymphocytic activity through SHP-1 signaling[24–29] and is expressed on CD8$^+$ T cells following integrin beta 2 (CD18) binding to collagen. Therapeutic inhibition of intratumoral collagen deposition through LOXL2 suppression[30,31] sensitizes resistant lung tumors to PD-L1 blockade. Furthermore, targeting LAIR1 signaling through LAIR2 overexpression[32] or SHP-1 inhibition sensitizes resistant tumors to PD-1 blockade and markedly reduces tumor growth and metastasis. Lastly, the analysis of melanoma patients treated with PD-1 blockade reveals that increasing gene expression of collagen, LAIR1, or TIM-3 predicts poorer overall survival or therapeutic response to immune checkpoint blockade. Our work identifies collagen and LAIR1 as a potential marker of PD-1/PD-L1 blockade resistance in lung cancer and validates multiple therapeutic targets in combination with immune checkpoint blockade.

## Results

**Anti-PD-1/PD-L1 resistant tumors have increased collagen.** To identify markers of PD-1/PD-L1 blockade resistance and recapitulate the unresponsiveness of late-stage disease to therapy, we subcutaneously implanted immunosuppressive 344SQ KP murine lung cancer cells with high levels of PD-L1[12] into syngeneic immunocompetent wild-type (WT) mice, and treated mice weekly with anti-PD-L1 antibody 7 days post-implantation, as previously described[8,12], or 21 days post-implantation when tumors were ~150–200 mm$^3$ in size (Fig. 1a). Tumors treated 1-week post-implantation showed an initial suppression of tumor growth, but eventually developed resistance to PD-L1 blockade, while tumors treated after 3 weeks were unresponsive to therapy (Fig. 1a). Reverse-phase protein array (RPPA) analysis[33,34] of resistant tumors that were treated 1-week post-implantation in conjunction with previous mRNA profiling from comparable experiments[8] revealed a consistent, statistically significant upregulation of multiple collagen isoforms in tumors that developed resistance to anti-PD-L1 blockade (Fig. 1b (RPPA) and c (RNA)). Because antibody validation requirements for RPPA limits the collagen isoforms that can be assessed on the arrays, we performed Masson's trichrome analysis of lung tumor tissues at 1 and 3 weeks post-implantation without treatment and observed higher levels of total collagen after 3 weeks of growth when tumors were innately unresponsive to treatment versus the 1-week samples (Fig. 1d). Additionally, validation of the RPPA and RNA profiling data by western blotting and trichrome staining showed increased intratumoral collagen deposition in the 1-week post-implantation-treatment lung tumor tissues after 7 weeks of treatment, at which point they displayed acquired resistance to PD-1 or PD-L1 blockade (Fig. 1e, Supplementary Fig. 1a and b). Despite the increase in collagen deposition, we did not observe an increase in LOXL2 expression following PD-L1 blockade (Supplementary Fig. 1a), as immunosuppressive KP tumors already express elevated levels of LOXL2[15].

We expanded our analysis into autochthonous tumors from $Kras^{LSL-G12D};p53^{fl/fl}$ (KP) mice that were induced with adenovirus expressing *Cre* recombinase and treated with PD-L1 or PD-1 blockade for 12–14 weeks when mice developed resistance to therapy[8]. Utilizing second harmonics generation (SHG) microscopy to more sensitively visualize lung tumor collagen fibers, we observed a consistent increase in lung tumor-associated collagen when KP mice were treated with either anti-PD-1 or -PD-L1 (Fig. 1f). Our findings demonstrate that (1) baseline tumor collagen deposition and (2) accelerated checkpoint blockade treatment-induced intratumoral collagen deposition correlates with de novo or acquired PD-L1/PD-1 axis blockade resistance, respectively.

**LOXL2 suppression decreases exhausted CD8$^+$ T cells.** We next sought to identify changes in the tumor immune microenvironment following a reduction in intratumoral collagen. Previous work demonstrated that LOXL2 knockdown or inhibition of LOXL2 enzymatic activity with ellagic acid diminished collagen deposition in our KP lung tumor models and thereby reduced metastasis[15,31]. Utilizing these reagents, we implanted metastatic, collagen-rich 344SQ KP cells with stable LOXL2 shRNA knockdown or treated mice containing wild-type tumors with ellagic acid LOXL2 inhibitor feed starting one-week post-implantation. We compared these tumors to immunogenic, non-metastatic, low-collagen 393P KP tumors[12,15,35] as a phenotypic control. After 5 weeks of tumor growth, tissues were processed and intratumoral immune populations were assessed by FACS analysis (Fig. 2a). Consistent with our prior findings[15,31], LOXL2 knockdown (Supplementary Fig. 2a, b) or enzymatic inhibition in

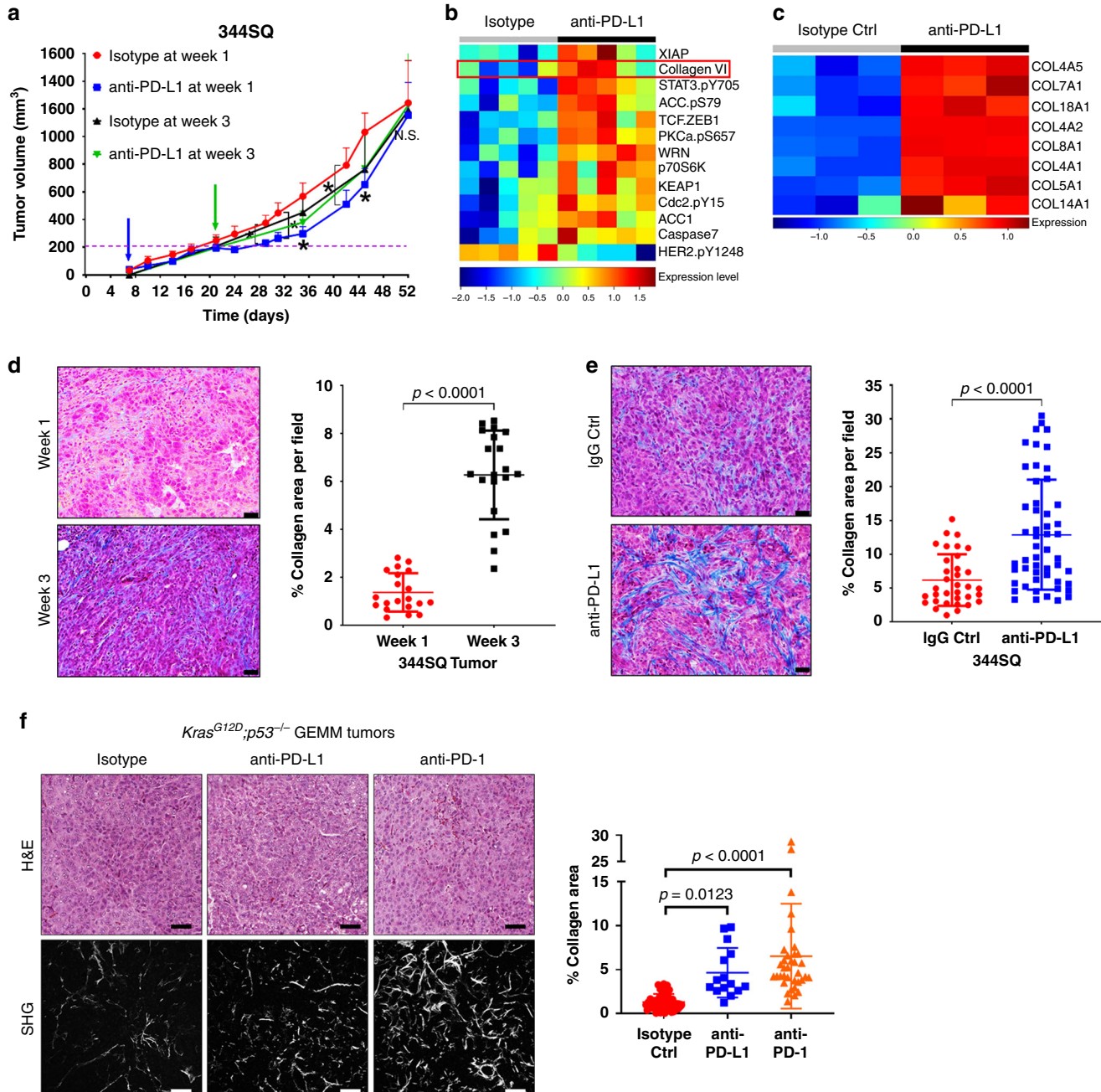

**Fig. 1 PD-(L)1 blockade is associated with increased intratumoral collagen deposition. a** In vivo volume measurements at indicated time points for 344SQ subcutaneous tumors implanted in syngeneic WT mice treated weekly with anti-PD-L1 (200 µg/mouse) or isotype control (200 µg/mouse). Treatment start time denoted by blue (1-week post-implantation) or green (3-week post-implantation) arrow; $n = 5$ mice per treatment group. Statistics calculated using a one-way repeated mixed-effects model (REML) with *$p = 0.0371$. **b** Heatmap of RPPA profile showing statistically significant (FDR < 0.05) differentially expressed proteins in 344SQ subcutaneous tumors treated weekly with anti-PD-L1 or isotype control, starting 1-week post-implantation in mice. Tumor samples were collected for RPPA at the endpoint of experiment in (**a**), ~7 weeks, when tumors developed resistance. **c** Heatmap of statistically significant (FDR < 0.05) differentially expressed mRNAs related to collagen genes in 344SQ tumors treated weekly with anti-PD-L1 or isotype control, starting after one week of implantation. Tumors samples and RNA profiling data were obtained from a prior study[8] at treatment endpoint, ~8 weeks, when tumors developed resistance. **d** Representative Masson's trichrome stains and quantification of percent collagen area per field of untreated 344SQ tumors collected at 1 or 3 weeks post-implantation in mice; $n = 5$ tumors per time point and four microscopy fields per tumor sample were analyzed. Scale bars, 50 µm. Statistics calculated using two-sided student's t-test. **e** Representative trichrome stains and quantification of percent collagen area per field of 344SQ tumors treated weekly with IgG isotype control or anti-PD-L1, starting after 1 week of implantation. Tumors were analyzed at endpoint of the experiment in (**a**); $n = 5$ tumors per treatment group with 34 total microscopy fields analyzed for IgG Ctrl and 52-total fields analyzed for anti-PD-L1 groups. Scale bars, 50 µm. Statistics calculated using a two-sided student's t-test. **f** Left: Representative H&E stains and SHG microscopy, including SHG quantification of percent collagen area per field of lung tumor tissues from $Kras^{G12D};p53^{-/-}$ (KP) mice treated weekly with anti-PD-1, -PD-L1 or isotype control for 12–14 weeks; $n = 5$ lung tissues per treatment group with 45 (isotype Ctrl), 15 (anti-PD-L1), and 35 (anti-PD-1) total microscopy fields analyzed across all tissues in respective groups. Scale bars, 100 µm. Data presented as mean +/− SD. Statistics calculated using one-way ANOVA post hoc Tukey for multiple comparisons. *$P < 0.05$.

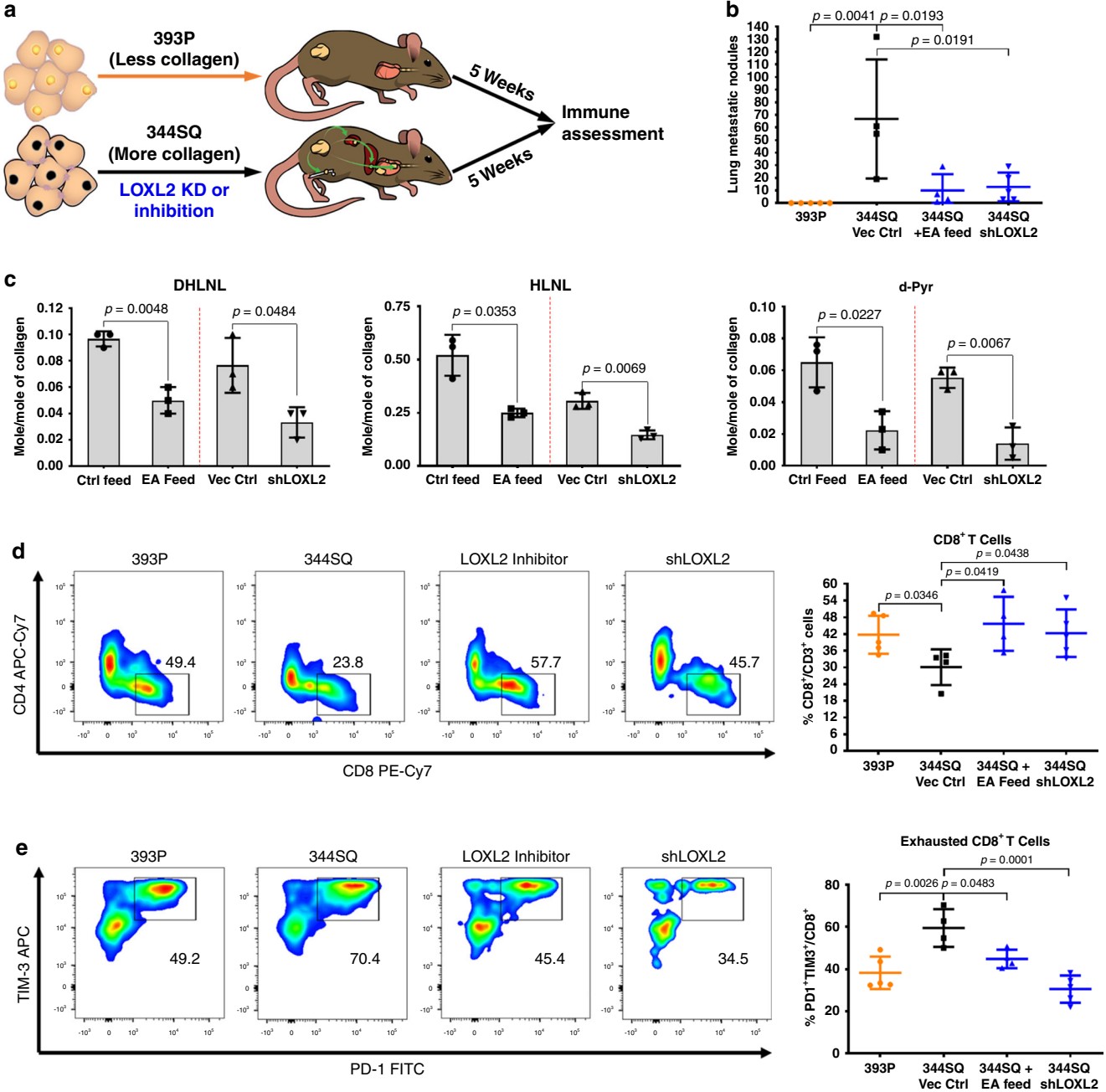

**Fig. 2 Tumors with decreased collagen have increased CD8⁺ TILs and decreased exhausted CD8⁺ TILs. a** Experimental schema illustrating immune assessment of 393P or 344SQ KP syngeneic tumors with LOXL2 knockdown or enzymatic inhibition. Tumors were collected and analyzed 5 weeks following subcutaneous implantation. Ellagic acid LOXL2 inhibitor was administered through mice feed 1-week following tumor implantation. **b** Quantification of lung metastatic surface nodules for indicated tumor groups after 5 weeks of tumor growth; 393P and 344SQ-shLOXL2 $n = 5$ mice per group; 344SQ Vec Ctrl and EA feed $n = 4$ mice per group. **c** Ratio of moles of DHLNL, HLNL, and d-Pyr collagen crosslinks per mole of total collagen analyzed by mass spectrometry in 344SQ tumors treated with EA feed LOXL2 inhibitor or LOXL2 shRNA; $n = 3$ tumors per treatment group. Data presented as mean +/− SD. Statistics calculated as a two-tailed student's $t$-test between different two groups from two separate experiments as denoted by a dashed-red line. **d** Left: Representative FACS plots of indicated tumor cell suspensions for percentage of CD8⁺ T cells gated from CD45⁺CD3⁺ cells. Right: Quantification of percentage of total CD8⁺ T cells for each individual tumor sample; 393P and 344SQ-shLOXL2 $n = 5$ mice per group; 344SQ Vec Ctrl and EA feed $n = 4$ mice per group. **e** Left: Representative FACS plots of indicated tumor cell suspensions for percentage of PD-1⁺TIM-3⁺ cells gated from CD8⁺ T cells in (**d**). Right: Quantification of percentage of PD-1⁺TIM-3⁺ CD8⁺ T cells for each individual tumor sample; 393P and 344SQ-shLOXL2 $n = 5$ mice per group; 344SQ Vec Ctrl and EA feed $n = 4$ mice per group. All data presented as mean +/− SD. Statistics calculated using a one-way ANOVA post hoc Tukey test.

344SQ tumors reduced metastatic disease (Fig. 2b), collagen crosslinking (Fig. 2c), and collagen deposition (Supplementary Fig. 2c and d) with no significant effect on primary tumor growth (Supplementary Fig. 2e). Flow cytometry analysis (sample flow

gating strategy of 344SQ tumors shown in Supplementary Fig. 3) revealed that LOXL2 inhibition or knockdown increased tumor-infiltrating CD8⁺ T cells and decreased PD-1⁺TIM-3⁺ exhausted CD8⁺ T cells (Fig. 2d and e), comparable to 393P tumors with

low baseline levels of collagen, with no significant effect on total CD3[+] T cells (Supplementary Fig. 4a). We further analyzed additional subpopulations of T cells and antigen-presenting cells (APCs), including total CD4[+] T cells, regulatory (CD25[+]FoxP3[+]) and ICOS CD4[+] T cells, memory/effector and naive CD8[+] T cells, F4/80[+] macrophages, CD11c[+] dendritic cells, and CD11b[+]GR1[+] myeloid-derived suppressor cells (MDSCs), and observed no significant or consistent changes in these immune cell subpopulations upon LOXL2 modulation (Supplementary Fig. 4b–d).

**LOXL2 suppression sensitizes lung tumors to PD-L1 blockade.** Previous work demonstrated that PD-(L)1 blockade transiently reduced tumor growth and metastasis in our syngeneic tumor models due to increased CD8[+] tumor-infiltrating lymphocytes (TILs) and decreased exhausted CD8[+] TILs[8,12,36]. Since increased intratumoral collagen is associated with treatment resistance to PD-L1 blockade, decreased CD8[+] TILs, and increased exhausted CD8[+] TILs, we next assessed whether the reduction in collagen deposition through LOXL2 suppression could sensitize resistant tumors to PD-L1 blockade. Either ellagic acid inhibition or shRNA knockdown of *LOXL2* was sufficient to decrease tumor collagen deposition and metastasis as previously observed[31], but primary tumor growth was reduced only when LOXL2 suppression was combined with PD-L1 blockade and was even able to do so when administered late in the course of tumor growth when tumors had reached ~200 mm³ (Fig. 3a and b, Supplementary Fig. 5a and b). Additionally, ellagic acid-mediated LOXL2 inhibition was more effective than tumor cell-intrinsic LOXL2 knockdown at reducing tumor growth when combined with anti-PD-L1, eliciting an initial reduction in tumor size that was sustained throughout the course of treatment (Fig. 3a vs. Supplementary Fig. 5a). FACS analysis of tumor tissues at the treatment endpoints showed that the combination of PD-L1 and LOXL2 inhibition significantly increased total CD8[+] TILs and decreased exhausted CD8[+] T cell subpopulations (Fig. 3c). Although LOXL2 knockdown did not increase total CD8[+] TILs at the treatment endpoint, LOXL2 knockdown in combination with PD-L1 blockade reduced the intratumoral exhausted CD8[+] TILs (Supplementary Fig. 5c).

To determine if the observed changes in intratumoral CD8[+] T cell populations following combination therapy were necessary for a reduction in tumor size, we treated 344SQ tumors with ellagic acid and anti-PD-1, along with control or a CD8-depleting antibody. Mice were treated with anti-PD-1 as PD-1 blockade has been shown to produce a higher percentage of durable response in lung cancer patients[4]. Similar to PD-L1 blockade, anti-PD-1 in combination with ellagic acid significantly reduced lung tumor growth and metastasis while CD8 depletion negated the effects of the combinatorial treatment (Fig. 3d and e). Consistently, FACS analysis of tumor tissues at the treatment endpoint showed a significant increase in total CD8[+] TILs and a decrease in exhausted CD8[+] T cell subpopulations with combination treatment (Fig. 3f). Because CD8 depletion markedly reduced total CD8[+] T cell numbers, we were unable to analyze CD8[+] T cell subpopulations in the anti-CD8 depletion group (Fig. 3f). Although our findings demonstrated that CD8[+] T cells were indispensable for combination treatment efficacy, the decrease in exhausted CD8[+] T cells from LOXL2 inhibition or knockdown did not fully characterize the observed phenotype as the addition of PD-(L)1 blockade was still required for tumor growth reduction. Thus, we analyzed intracellular IFN-γ and IL-2 cytokine levels in total CD8[+] TILs by FACS and observed an increase in IFN-γ[+] and a decrease in IL-2[+] CD8[+] TILs following combination therapy, suggesting that CD8[+] TILs are in an activated or effector state while IL-2 is potentially secreted by

other immune cell types to stimulate CD8[+] T cell differentiation and activation (Fig. 3f, Supplementary Fig. 6a).

To further confirm that CD8[+] T cells infiltrate into tumor tissues after combination therapy with anti-PD-L1 and LOXL2 inhibition or knockdown, we performed immunohistochemistry (IHC) stains for CD8 in primary and lung metastatic tumor tissues from the combinatorial treatment studies (Fig. 3a and Supplementary Fig. 5a). Histological analyses verified that LOXL2 suppression decreased lung metastatic burden and increased CD8 cell infiltration into both the primary and metastatic tumors (Fig. 3g and h, Supplementary Fig. 6b–e). Additionally, tumor-associated CD8 cell numbers were comparable between the peripheral and inner regions of primary tumors (Fig. 3g), but were further increased in primary and metastatic tumors relative to total tumor cells when LOXL2 inhibition or knockdown was combined with anti-PD-L1 (Fig. 3g and h, Supplementary Fig. 6b–e).

Lastly, to extend our findings beyond KP tumor models, we tested the combinatorial treatment studies in a second, previously described, syngeneic Lewis lung cancer (LLC-JSP) model[12]. We confirmed that LLC-JSP cells express higher levels of collagen and LOXL2 even when compared to the metastatic, high-collagen 344SQ KP murine cell line model (Supplementary Fig. 7a). Consistently, ellagic acid inhibition of LOXL2 in combination with anti-PD-L1 significantly reduced tumor volume and weight (Supplementary Fig. 7b–d). Furthermore, LLC-JSP tumors treated with the combinatorial therapy were unable to form solid, palpable tumors compared to untreated or monotherapy controls (Supplementary Fig. 7e). Because of the non-palpable tumors at the experimental endpoint, we were unable to perform FACS analysis on LLC-JSP tumors as the combinatorial treatment group did not yield sufficient viable cells for acceptable quantification. Our findings demonstrate that inhibition of collagen deposition by targeting LOXL2 can overcome anti-PD-(L)1 resistance by increasing the infiltration and activation state of intratumoral CD8[+] T cells in multiple syngeneic models of lung cancer.

**CD18 induces LAIR1/SHP-1-dependent CD8[+] T cell exhaustion.** Next, we sought to explore the mechanism of collagen-induced CD8[+] T cell exhaustion and identify potential therapeutic interventions. Previous reports have implicated the collagen-binding leukocyte associated immunoglobulin-like receptor 1 (LAIR1) in suppressing T cell, NK cell, and dendritic cell activity through the inhibitory protein tyrosine phosphatase SHP-1 pathway[24–29,37,38] (Fig. 4a). To date, there exists no commercial inhibitory antibodies or small-molecule inhibitors that specifically target LAIR1. In order to study the role of LAIR1 and SHP-1 in T cell exhaustion, we performed in vitro co-culture assays with anti-CD3/CD28 T cell activated splenocytes in suspension on plastic culture dishes or in the presence of collagen. These assays were performed with control or treatment with the SHP-1 inhibitor, TPI-1[39], the SHP099 SHP-2 inhibitor[40], or conditioned media containing LAIR2, the secreted homolog of LAIR1 that binds competitively to the shared collagen epitope[32]. FACS analyses for the CD8[+] T cell subpopulations were conducted at the experimental endpoint (Fig. 4b).

First, to determine if collagen specifically induces T cell exhaustion rather than T cell interaction with ECM in general, splenocytes were co-cultured on plastic alone or with laminin-rich ECM (Matrigel) or collagen, which demonstrated that only collagen decreased total CD8[+] T cells and induced LAIR1 expression and PD-1[+]TIM-3[+] exhaustion in CD8[+] T cells (Supplementary Fig. 8a). Inhibition of SHP-1 or SHP-2 had no effect on splenocyte viability, total CD8[+] T cells, or LAIR1

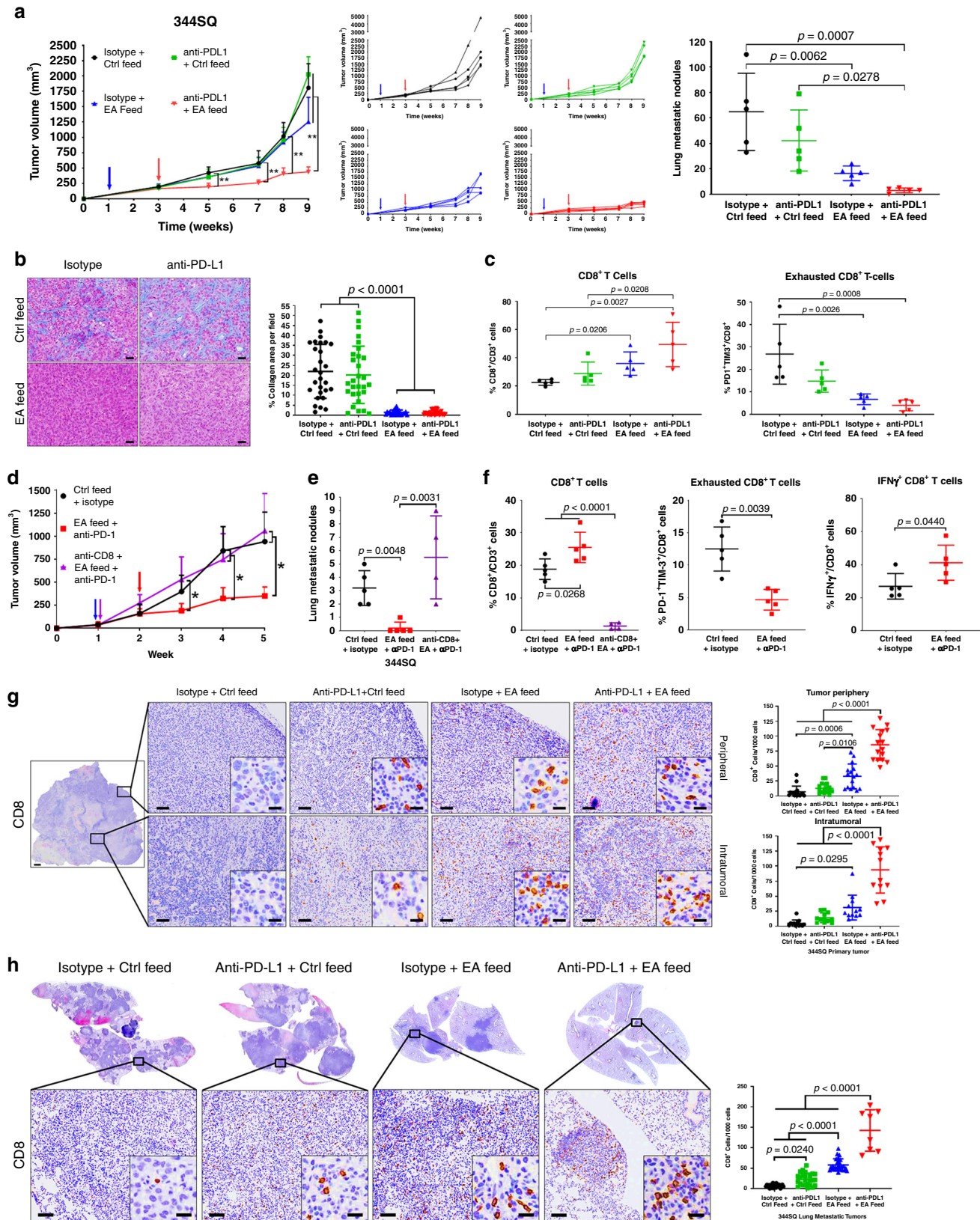

expression but did significantly ($P < 0.05$) decrease PD-1+TIM-3+ exhausted CD8+ T cells when co-cultured with collagen (Fig. 4c, Supplementary Fig. 8b). To study the effect of LAIR2 on LAIR1-induced CD8+ T cell exhaustion, we generated 344SQ cells that constitutively express human LAIR2, as mice do not possess the LAIR2 gene[26,41]. LAIR2 secretion into the conditioned media was

validated by ELISA (Supplementary Fig. 8c) and splenocytes were co-cultured in LAIR2-conditioned media in the absence or presence of collagen. LAIR2 reduced collagen-induced CD8+ T cell exhaustion, with no effect on total CD8+ T cells, and modestly, but significantly ($P = 0.017$) decreased collagen-induced LAIR1 expression on CD8+ T cells (Fig. 4d).

**Fig. 3 Reduction in LOXL2-dependent collagen deposition sensitizes tumors to PD-L1/PD-1 blockade. a** Left: Tumor volume measurements at indicated time points for 344SQ subcutaneous tumors treated with ellagic acid (EA) feed, single-agent anti-PD-L1 (200 µg/mouse/week), or both drugs in combination. Starting time of ellagic acid feed indicated by the blue arrow. Starting time of PD-L1 blockade denoted by the red arrow. Middle: Tumor volume measurements for individual mice in each treatment group. Right: Quantification of lung metastatic surface nodules (right) in indicated treatment groups at endpoint of the experiment; $n = 5$ mice per treatment group; $**p < 0.01$. **b** Representative trichrome stains and quantification of percent collagen area per field of 344SQ tumors in the indicated treatment groups at the endpoint of the experiment from (**a**); $n = 5$ tumors per treatment group with 28 (isotype + Ctrl feed), 29 (anti-PD-L1 + Ctrl feed), 23 (isotype + EA feed), and 30 (anti-PD-L1 + EA feed) total fields analyzed across all tissues in respective groups. Scale bars, 50 µm. **c** Left: FACS quantification of percentage of total CD8$^+$ T cells gated from CD45$^+$CD3$^+$ cells for each individual tumor sample from the experiment in (**a**). Right: FACS quantification of percentage of PD-1$^+$TIM-3$^+$ cells gated from total CD8$^+$ T cells for each individual tumor sample; $n = 5$ tumors per group. **d** Tumor volume measurements at indicated time points for 344SQ tumors treated with EA feed and anti-PD-1 (200 µg/mouse/week) with or without antibody depletion of CD8 T cells (200 µg/mouse/week); $n = 5$ tumors per group. Starting time of EA feed and anti-CD8 depletion marked by the blue and purple arrow. Red arrow denotes the starting time of PD-L1 blockade; $*p < 0.05$. **e** Quantification of lung metastatic surface nodules in indicated treatment groups at the endpoint of the experiment from (**d**). **f** FACS quantification of percentage of total CD8$^+$ (left), PD-1$^+$TIM-3$^+$ CD8$^+$ (middle), and IFN-γ$^+$ CD8$^+$ (right) TILs in tumor cell suspensions at experimental endpoint from (**d**). Statistics calculated using one-way ANOVA post hoc Tukey test for multi-group or two-tailed student's *t*-test for two-group comparisons. **g** Representative CD8 IHC stains with quantification of peripheral and intratumoral primary tumor regions as denoted by black zoom box in 344SQ tumors from the experiment in (**a**); $n = 5$ tumors per group. For the tumor periphery, 16 total fields were analyzed across all tumors for each treatment group. For intratumoral regions, 12 total fields were analyzed across all tumors for each treatment group. Scale bars, 100 µm. Inset scale bars, 20 µm. **h** Representative CD8 IHC stains with quantification of metastatic lung tumor regions as denoted by black zoom box from the experiment in (**a**); $n = 5$ lungs for isotype + Ctrl feed, anti-PD-L1 + Ctrl feed, isotype + EA feed treatment groups, with five tumor fields quantified per sample; $n = 4$ lungs for anti-PD-L1 + EA feed group with two fields quantified per sample Scale bars, 100 µm. Inset scale bars, 20 µm. All data presented as mean +/− SD. Unless stated, statistics calculated using a one-way ANOVA post hoc Tukey test.

To determine if lung cancer cells directly induce LAIR1 expression and CD8$^+$ T cell exhaustion due to collagen secretion as opposed to confounding secondary factors in vivo, splenocytes were co-cultured in vitro with 344SQ cells following LOXL2 knockdown. Co-culture with 344SQ cells reduced total CD8$^+$ T cells, induced LAIR1 expression, and increased CD8$^+$ T cell exhaustion, which were restored to comparable levels as control splenocytes when co-cultured with LOXL2 knockdown cells (Supplementary Fig. 9a–c). Analysis of 344SQ tumors following LOXL2 suppression in combination with PD-L1 blockade from the prior experiments (Fig. 3a, Supplementary Fig. 6) demonstrated that LAIR1 expression on CD8$^+$ TILs was reduced in 344SQ tumors following LOXL2 knockdown or inhibition (Fig. 4e and f). Interestingly, LAIR1$^+$CD8$^+$ TILs increased when LOXL2 knockdown tumors were treated with PD-L1 blockade, consistent with the corresponding increase in total collagen levels (Fig. 4e, Supplementary Fig. 5b). Finally, splenocyte co-culture with 393P or 344SQ KP cells treated with SHP-1 or SHP-2 inhibitors demonstrated that only SHP-1 inhibition consistently reduced tumor cell-induced exhausted CD8$^+$ T cells (Supplementary Fig. 9d).

Although our findings showed that CD8$^+$ T cell interaction with collagen-induced LAIR1 expression, abrogating LAIR1 signaling through LAIR2 competitive binding or SHP-1 inhibition did not show a significant or biologically relevant decrease in LAIR1 expression (Fig. 4c and d), suggesting that LAIR1 expression in T cells is induced by an alternative collagen receptor. To identify potential receptors, we performed an unbiased correlation analysis between *LAIR1* mRNA expression and all genes sampled in the TCGA LUAD and LUSC patient datasets. TCGA correlation analysis identified the leukocyte-specific integrin beta 2 (CD18) collagen receptor as having the highest correlation value to LAIR1 along with the CD18 alpha subunit binding partners *ITGAL* (CD11a), *ITGAM* (CD11b), and *ITGAX* (CD11c) (Fig. 4g and h, Supplementary Fig. 8e and f). Although the collagen-like C1q family of genes also showed a positive correlation with LAIR1, prior studies have established that the complement system is involved in regulating LAIR1 activity and signaling rather than LAIR1 expression[42].

To validate whether CD18 induces LAIR1 expression following leukocyte-collagen interaction, we co-cultured anti-CD3/CD28 activated splenocytes in vitro on plastic, Matrigel, or Matrigel/collagen and treated splenocytes with a CD18 neutralizing antibody. Consistent with our prior results, collagen decreased total CD8$^+$ T cells and increased LAIR1$^+$ and PD-1$^+$TIM-3$^+$ exhausted CD8$^+$ T cell subpopulations (Fig. 4i). However, inhibition of CD18 markedly reduced collagen-induced LAIR1 expression and CD8$^+$ T cell exhaustion down to comparable levels as T cells cultured on plastic (Fig. 4j and k). Inhibition of CD18 showed no significant effect on total CD8$^+$ T cell numbers and did not affect LAIR1$^+$ and PD-1$^+$TIM-3$^+$ exhausted CD8$^+$ T cell levels when cultured on Matrigel, suggesting that CD18 binding and activity is specific to collagen. Our in vitro co-culture studies identify a regulatory loop whereby tumor-associated collagen recognition by CD18 (ITGβ2) on leukocytes upregulates LAIR1 expression, which subsequently promotes CD8$^+$ T cell exhaustion through SHP-1 signaling.

**LAIR2 expression sensitizes lung tumors to PD-1 blockade.** Because tumor-associated collagen induces CD8$^+$ T cell exhaustion through LAIR1-SHP-1 signaling, we next sought to assess LAIR1 and SHP-1 as potential therapeutic targets in combination with immune checkpoint blockade. Because there are no available inhibitors or antibodies that target LAIR1, we implanted 344SQ cells that constitutively express secreted LAIR2 to inhibit LAIR1 binding to collagen, and treated mice with anti-PD-1 blocking antibody starting 3 weeks post-implantation. Although LAIR2 expression alone slowed tumor growth modestly, lung tumor growth and metastasis were significantly reduced only when LAIR2 overexpression was combined with anti-PD-1 treatment (Fig. 5a and b, Supplementary Fig. 10a). Reduction in lung tumor growth was observed within 1 week of treatment and was sustained throughout the course of treatment (Fig. 5a). FACS analysis of tumor tissues showed that LAIR2 overexpression was sufficient to increase total CD8$^+$ TILs and reduce exhausted CD8$^+$ with no effect on LAIR1$^+$CD8$^+$ subpopulations (Supplementary Fig. 10b). The addition of PD-1 blockade with LAIR2 overexpression showed a decrease in exhausted CD8$^+$ T cell numbers and an increase in total CD8$^+$ and effector CD8$^+$ TILs (Fig. 5c), consistent with reports that downstream SHP-1 signaling regulates multiple lymphocyte functions including memory/effector and CD69 T cell activation in a context-dependent manner[27,43,44].

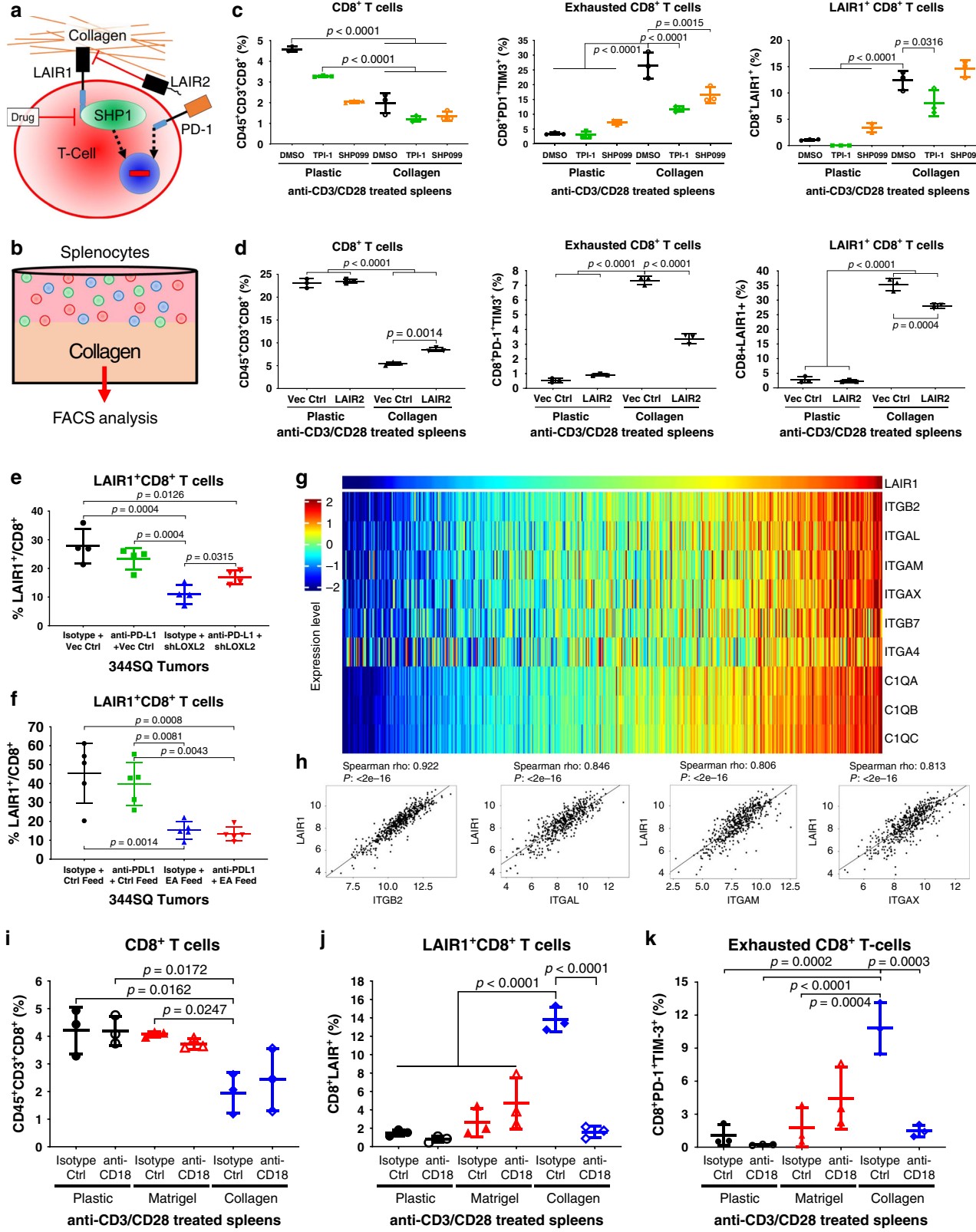

Histologic analyses of primary and metastatic lung tumor tissues showed an increase in CD8 T cell infiltration with no reduction in collagen deposition when LAIR2 was constitutively expressed, while the addition of PD-1 blockade further increased CD8 T cell infiltration in primary and metastatic tumors with no effect on collagen deposition (Fig. 5d and e, Supplementary Fig. 10c).

**SHP-1 inhibition sensitizes lung tumors to PD-1 blockade**. To validate the LAIR1 downstream effector molecule SHP-1 as a therapeutic target, 344SQ tumors were treated with the SHP-1 inhibitor, TPI-1, which was sufficient as a single-agent to reduce primary tumor growth (Fig. 6a, Supplementary Fig. 10d). Although the combination of PD-1 blockade with SHP-1

**Fig. 4 Collagen induces CD8+ T cell exhaustion through LAIR1-mediated SHP-1 signaling. a** Proposed model illustrating LAIR1 binding to collagen, which induces SHP-1-mediated T cell exhaustion alternative to the PD-1 pathway. LAIR1 signaling can be inhibited by secreted LAIR2 homolog binding to the LAIR1 collagen epitope or small-molecule inhibition of SHP-1. **b** Schematic of in vitro co-culture experiments. Splenocytes isolated from WT mice were co-cultured on collagen versus on plastic as a control in LAIR2-conditioned media, media containing SHP-1 inhibitor (TPI-1) or SHP-2 inhibitor (SHP099), and then analyzed by FACS. **c** FACS percentage of total CD8+ (left), PD-1+TIM-3+ CD8+ (middle), and LAIR1+ CD8+ (bottom) T cells in splenocytes co-cultured on plastic or collagen treated with 10μM SHP-1 inhibitor (TPI-1), 20 μM SHP-2 inhibitor (SHP099), or DMSO control for 96 h; $n = 3$ technical triplicates. **d** FACS percentage of total CD8+ (left), PD-1+TIM-3+ CD8+ (middle), and LAIR1+ CD8+ (right) T cells in splenocytes co-cultured on plastic or collagen for 96 h in conditioned media from 344SQ cells expressing LAIR2 or vector control; $n = 3$ technical triplicates. **e** FACS percentage of LAIR1+ CD8+ T cells for indicated 344SQ tumor cell suspensions from the experiment in Supplementary Fig. 5; $n = 4$ tumors per group. **f** FACS percentage of LAIR1+ CD8+ T cells for indicated 344SQ tumor cell suspensions from the experiment in Fig. 3; $n = 5$ tumors per group. **g** Heatmap showing association of mRNA expression in TCGA LUSC dataset between LAIR1 and collagen receptor genes that are statistically significant ($P < 0.05$) by Spearman's rank correlation. **h** Representative scatter plot analysis of Spearman's rank correlation between ITGB2, ITGAL, ITGAM, and ITGAX versus LAIR1 mRNA expression in TCGA LUSC dataset. **i–k** Splenocytes cultured in vitro on plastic, laminin-rich Matrigel, or collagen were treated with CD18 inhibitory antibody (10 μg/mL) for 96 h and analyzed by FACS for percentage of (**i**) total CD8+ T cells gated from CD45+CD3+ splenocytes, (**j**) LAIR1+ CD8+ T cells, and (**k**) PD-1+TIM-3+ exhausted CD8+ T cells; $n = 3$ technical replicates per group. Unless stated, all data presented as mean $+/-$ SD and statistics calculated using a one-way ANOVA post hoc Tukey test.

inhibition presented no additional reduction in primary tumor growth compared to SHP-1 inhibition alone, the combinatorial treatment significantly ($P < 0.01$) reduced metastatic disease (Fig. 6b). Despite the increasing trend with SHP-1 inhibition alone, CD8+ TILs only showed a significant increase when combined with anti-PD-1 (Fig. 6c). However, additional FACS analyses confirmed a decrease in exhausted CD8+ and an increase in activated CD69+ CD8+ TILs when tumors were treated with SHP-1 inhibitor as a single agent or in combination with PD-1 blockade (Fig. 6c). Histologic analyses of primary tumor tissues consistently showed that TPI-1 treatment alone increased CD8 infiltration, which was further increased when anti-PD-1 was combined with SHP-1 inhibition (Fig. 6d) with no observed effect on collagen deposition in either treatment group (Supplementary Fig. 10f). Interestingly, histologic analyses of metastatic lung tumors only showed a large increase in CD8 infiltration when TPI-1 was combined with anti-PD-1 (Fig. 6e), consistent with the observed changes in metastatic tumor growth.

To verify that SHP-1 or -2 inhibition promotes immune-mediated, rather than tumor cell-intrinsic, tumor cell elimination, anti-CD3/CD28 activated splenocytes were co-cultured in vitro with 393P or 344SQ KP lung cancer cells in the presence or absence of TPI-1 or SHP099 at increasing concentrations. Following treatment, media containing splenocytes in suspension were removed and the relative tumor cell number was quantified. Controls with only tumor cells and the SHP-1/2 inhibitors showed no direct effects of the inhibitors on tumor growth at physiological dosages, compared to untreated controls (Fig. 6f). Co-culture experiments demonstrated that SHP-1 inhibition, but not SHP-2, reduced tumor cell numbers only in the presence of splenocytes in 393P (TPI-1 IC$_{50}$ = 72.24 μM; SHP099 IC$_{50}$ = 94.13 μM; Spleens + TPI-1 IC$_{50}$ = 20.37 μM; Spleens + SHP099 IC$_{50}$ = 60.47 μM, Fig. 6f) and 344SQ cells (TPI-1 IC$_{50}$ = 86.72 μM; SHP099 IC$_{50}$ = 101.2 μM; Spleens + TPI-1 IC$_{50}$ = 22.22 μM; Spleens + SHP099 IC$_{50}$ = 59.15 μM, Fig. 6f). Splenocytes co-cultured with tumor cells showed no significant effect on tumor cell numbers in the absence of SHP inhibitors (Supplementary Fig. 10g). Our findings validate LAIR1 and SHP-1 as potential combinatorial therapeutic targets to overcome PD-1/PD-L1 blockade resistance mediated by collagen deposition.

**Pre-clinical results observed in lung cancer patient samples**. To evaluate the clinical relevance of our findings, we pathologically assessed a cohort of treatment-naive patient lung tumor samples by tissue microarray (TMA) ($n = 451$) and whole sections ($n = 8–10$) for collagen I, collagen III, and multiple immune check-point markers by IHC from the previous studies[15,45,46]. TMA

analysis showed an inverse correlation ($r = -0.362$, $p < 0.0001$) between collagen area (collagen I and III) and CD8 T cell marker count, and a positive correlation ($r = 0.3838$, $p < 0.0001$) between collagen area and the TIM-3 exhaustion marker count (Fig. 7a and b). To address the limited tissue coverage by TMAs, we scored and averaged collagen area, CD8 and TIM-3 marker counts from five areas of whole sections of lung tumor tissues encompassing the inner and peripheral tumor regions, and observed a stronger inverse correlation between collagen I ($r = -0.693$, $p = 0.0308$) and collagen III ($r = -0.7416$, $P = 0.0178$) area versus CD8 T cell marker count (Fig. 7c). Conversely, we observed a positive correlation between collagen I ($r = 0.8246$, $P = 0.0158$) and collagen III ($r = 0.7545$, $P = 0.038$) area versus TIM-3 exhaustion marker count (Fig. 7a–c).

Next, we expanded our analyses to The Cancer Genome Atlas (TCGA) lung adenocarcinoma (LUAD) and squamous cell carcinoma (LUSC) patient datasets. We observed a positive correlation in mRNA expression between numerous collagen isoforms versus *LAIR1* or *HAVCR2* (TIM-3) in LUAD and LUSC patient samples (Fig. 7d–f, Supplementary Fig. 11a–c). Additionally, we observed a strong correlation between *LAIR1* and TIM-3 mRNA expression in LUAD ($r = 0.940$, $P < 2e-16$) and LUSC ($r = 0.954$, $P < 2e-16$) samples (Fig. 7g, Supplementary Fig. 11d). We next sought to explore our findings in the context of mutant *EGFR* lung cancers, which possess an immunosuppressive microenvironment but present with low levels of PD-L1 compared to *KRAS* mutant lung cancers[13]. However, comparison of collagen isoform mRNA expression between mutant *EGFR* and *KRAS* LUAD and LUSC TCGA samples showed no significant difference in collagen gene expression between the mutant subtypes (Supplementary Fig. 11e and f), suggesting another mechanism of immune suppression in mutant *EGFR* lung cancers. Patient tumor analyses corroborate our preclinical observations and identify collagen and LAIR1 as potential markers of immune suppression in lung cancers.

**Collagen, LAIR1, and TIM3 predict response to anti-PD-1**. To further extend the clinical relevance of our study, we sought to apply our findings to predict patient response to anti-PD-1/PD-L1 therapies. Because available datasets for lung cancer patients treated with immune checkpoint blockade do not have matched pre- and post-treatment samples, we analyzed publically available, published datasets from melanoma patients treated with anti-PD-1 therapies[47,48]. Gene expression analysis of multiple collagen isoforms in pre-treatment melanoma patient samples[47] revealed that upregulation of collagen mRNA correlated with progressive disease, while lower collagen mRNA correlated with a partial or

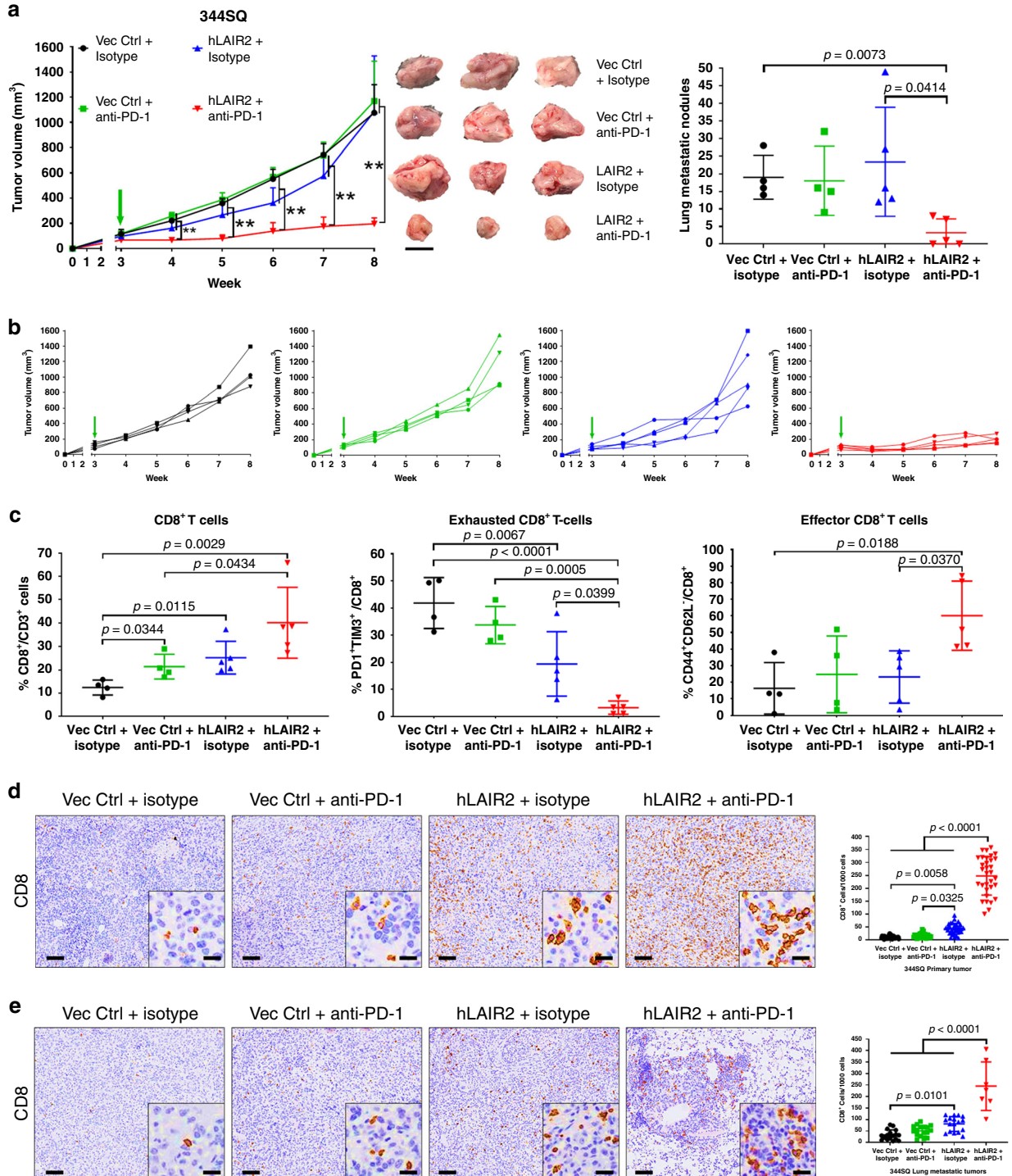

complete response when patients received anti-PD-1 therapies (Fig. 8a and b). Analysis of a separate dataset[48] comparing matched pre- and on-treatment melanoma patient samples showed a significant overall increase in *LAIR1* and *HAVCR2* (TIM-3) expression when patients received anti-PD-1 therapy (Fig. 8c), consistent with our experimental models. Although stratification of pre- and on-treatment patient samples by collagen, *LAIR1*, and *HAVCR2* expression levels did not predict significant changes in overall survival (Supplementary Fig. 11g and h), net changes in *LAIR1* and *HAVCR2* expression (defined as the difference in mRNA levels between pre- and on-treatment samples) predicted significant improvement in overall survival for

patients who showed a decrease in *LAIR1* and *HAVCR2* mRNA levels rather than an increase in levels while on treatment (Fig. 8d). Our findings from datasets of melanoma patients treated with anti-PD-1 therapies corroborate our preclinical lung cancer observations and validate collagen and LAIR1 as potential predictive markers of adaptive resistance to immunotherapy.

## Discussion

The failure of anti-PD-1/PD-L1 immunotherapies to sustain durable response in many lung cancer patients emphasizes the need to delineate tumor cell-intrinsic and tumor microenvironment resistance mechanisms, characterize biomarkers for patient

**Fig. 5 Combination of PD-1 blockade with LAIR2 overexpression reduces lung tumor growth and metastasis. a** Left: Tumor volume measurements at indicated time points for 344SQ syngeneic tumors ± LAIR2 overexpression treated weekly with anti-PD-1 (200 μg/mouse) or isotype control (200 μg/mouse). Starting time of PD-1 blockade denoted by green arrow. Middle: Images of three representative 344SQ primary subcutaneous tumors for each of the indicated treatment groups. Right: Quantification of lung metastatic surface nodules in indicated treatment groups at endpoint of the experiment. For Vec Ctrl + isotype and Vec Ctrl + anti-PD-1 n = 4 mice per treatment group. For hLAIR2 + isotype and hLAIR2 + anti-PD-1 n = 5 mice per treatment group. Scale bar, 1 cm. Statistics calculated using one-way ANOVA; **p < 0.01. **b** Tumor volume measurements for individual mice in each treatment group from the experiment in (**a**). **c** FACS quantification for percentage of total CD8+ (left), PD-1+TIM-3+ exhausted CD8+ (middle), and CD44+CD62L- effector (right) CD8+ TILs in indicated 344SQ tumor cell suspensions from the experiment in (a). **d** Representative CD8 IHC stains with quantification of 344SQ primary tumors from the experiment in (**a**); n = 4 tumors for Vec Ctrl + isotype and Vec Ctrl + anti-PD-1 groups; n = 5 tumors for hLAIR2 + isotype and hLAIR2 + anti-PD-1 groups, 32 total fields quantified across all tumor samples for each treatment group. Scale bars, 100μm. Inset scale bars, 20 μm. **e** Representative CD8 IHC stains with quantification of metastatic lung tumors from the experiment in (**a**); n = 4 lungs for Vec Ctrl + isotype, Vec Ctrl + anti-PD-1, and hLAIR2 + isotype groups, 17 total tumor fields quantified across all samples for each group; n = 3 lung for hLAIR2 + anti-PD-1 group with seven total tumor fields quantified across all samples. Scale bars, 100 μm. Inset scale bars, 20 μm. All data presented as mean +/− SD. Statistics calculated using a one-way ANOVA post hoc Tukey test.

selection, and develop rational combinatorial therapeutic strategies. Utilizing well-established, murine lung cancer models, we identify intratumoral collagen as a major source of immune suppression and PD-1/PD-L1 axis blockade resistance and confirm our findings in multiple lung cancer patient datasets. Since previous studies have demonstrated that collagen can be induced through TGF-β signaling[15,31,49], our findings that collagen downregulates cytotoxic T cell activity corroborate with, and mechanistically elaborate upon earlier reports that correlate TGF-β-associated genes with immune suppression and anti-PD-(L)1 resistance[18,19]. Thus, collagen serves as a potential marker to predict and target to enhance response to immunotherapies.

Mechanistically, we identify LAIR1 as the leukocyte-specific collagen receptor that suppresses CD8+ T cell activity through SHP-1 signaling and promotes PD-1/PD-L1 blockade resistance. Although the immunosuppressive function of LAIR1 is known, LAIR1's role and regulation in solid cancers is poorly defined. Our findings demonstrate that LAIR1 expression is induced following CD18 interaction specifically with collagen, further characterizing cancer's ability to commandeer normal wound healing processes and tissue-based immune homeostasis to evade destruction[50–53]. Since CD18 is ubiquitously expressed on leukocytes and is involved in adhesion and extravasation[54,55], our tumor FACS and histology data suggest that T cell recruitment is unimpeded in the collagen-rich tumor microenvironment, consistent with observations from prior studies[56–58], but in contrast to findings which suggest that collagen mechanically hinders T cell infiltration[18,59,60]. However, in vitro co-culture experiments show that immune cell interaction with collagen suppresses T cell cytotoxic function by decreasing total CD8+ T cell numbers and promoting a TIM-3+ exhausted CD8+ T cell phenotype. While inhibition of LAIR1 signaling or expression rescued CD8+ T cells from a collagen-induced exhaustion state, total CD8+ T cell numbers remained unaffected, suggesting that alternative collagen receptors and pathways are responsible for regulating CD3+ T cell differentiation into cytotoxic CD8+ T cells. Our mechanistic data are further strengthened by analyses of multiple patient datasets, which confirm the experimental observations that collagen and LAIR1 promote T cell exhaustion, and validate LAIR1 as an additional potential marker of immunotherapy sensitivity/resistance.

Therapeutically, because LOXL2 crosslinking is necessary for insoluble, mature collagen deposition in tissues[15,31,61], suppression of LOXL2 through genetic knockdown or pharmacologic inhibition decreased intratumoral collagen levels and sensitized lung tumors to PD-L1 blockade. Compared to LOXL2 knockdown, systemic LOXL2 inhibition by ellagic acid treatment was more effective at reducing tumor growth and did

not exhibit increased intratumoral collagen nor increased LAIR1+CD8+ TILs when combined with anti-PD-L1, which suggests that the source of collagen in tumors can originate from multiple cell types (e.g., cancer-associated fibroblasts) in addition to cancer cells[50]. Additionally, ellagic acid-binding to LOXL2's catalytic domain results in a metabolite byproduct that also inhibits TGF-β receptor I[31], imparting a secondary effect that reduces TGF-β-induced collagen production by cancer cells and prevents TGF-β recruitment and maintenance of regulatory T cells that are responsible for tumor immune suppression[62]. The disparity in response to anti-PD-L1 at varying stages of lung tumor growth underscores the importance of collagen in immunotherapy resistance and validates LOXL2 as a promising therapeutic target to combine with PD-1/PD-L1 blockade.

Although reducing intratumoral collagen sensitized lung tumors to PD-L1 blockade even at later stages of tumor formation, LOXL2 inhibition was required immediately following tumor implantation to prevent baseline collagen deposition. Unfortunately, the majority of lung cancer patients are diagnosed with late-stage disease, and histopathologic analyses show that lung tumors exhibit collagen deposition even at early stages of lung cancer[1,15]. Furthermore, LOXL2 is only involved in crosslinking and deposition of fibrillar collagen—the predominant collagen type in our metastatic lung cancer models[15]—but is not involved in the maturation of non-fibrillar collagens[63], which are also present in lung cancers and involved in epithelial tissue immune regulation[52]. Therefore, mechanistically validating LAIR1 and SHP-1 as additional therapeutic targets in combination with PD-1/PD-L1 blockade is vital to circumvent the need for early LOXL2 inhibition and to allow treatment of lung cancer patients with late-stage disease. Due to the lack of direct LAIR1 inhibitors to date, we utilized LAIR2 overexpression and SHP-1 inhibition to abrogate LAIR1 activity. Although LAIR2 overexpression and SHP-1 inhibition decreased exhausted CD8+ TILs, reduction in primary tumor growth and lung metastasis required combination with PD-1 blockade, suggesting that CD8+ T cell induction into an activated or effector state is also necessary for immune-mediated tumor cell elimination. Furthermore, SHP-1 inhibition alone reduced primary tumor growth but not metastasis, which suggests that SHP-1 regulates multiple aspects of T cell functions in addition to T cell exhaustion, and is independent of PD-1-regulated T cell biology in our system. Moreover, the TPI-1 inhibitor may potentially have off-target effects at higher doses[39], emphasizing the necessity to develop more specific therapeutic SHP-1 inhibitors. Finally, while inhibition of CD18 in vitro reduced LAIR1 expression and CD8+ T cell exhaustion, CD18 inhibitors have been ineffective and lethal in human patients due to the necessity of CD18 in

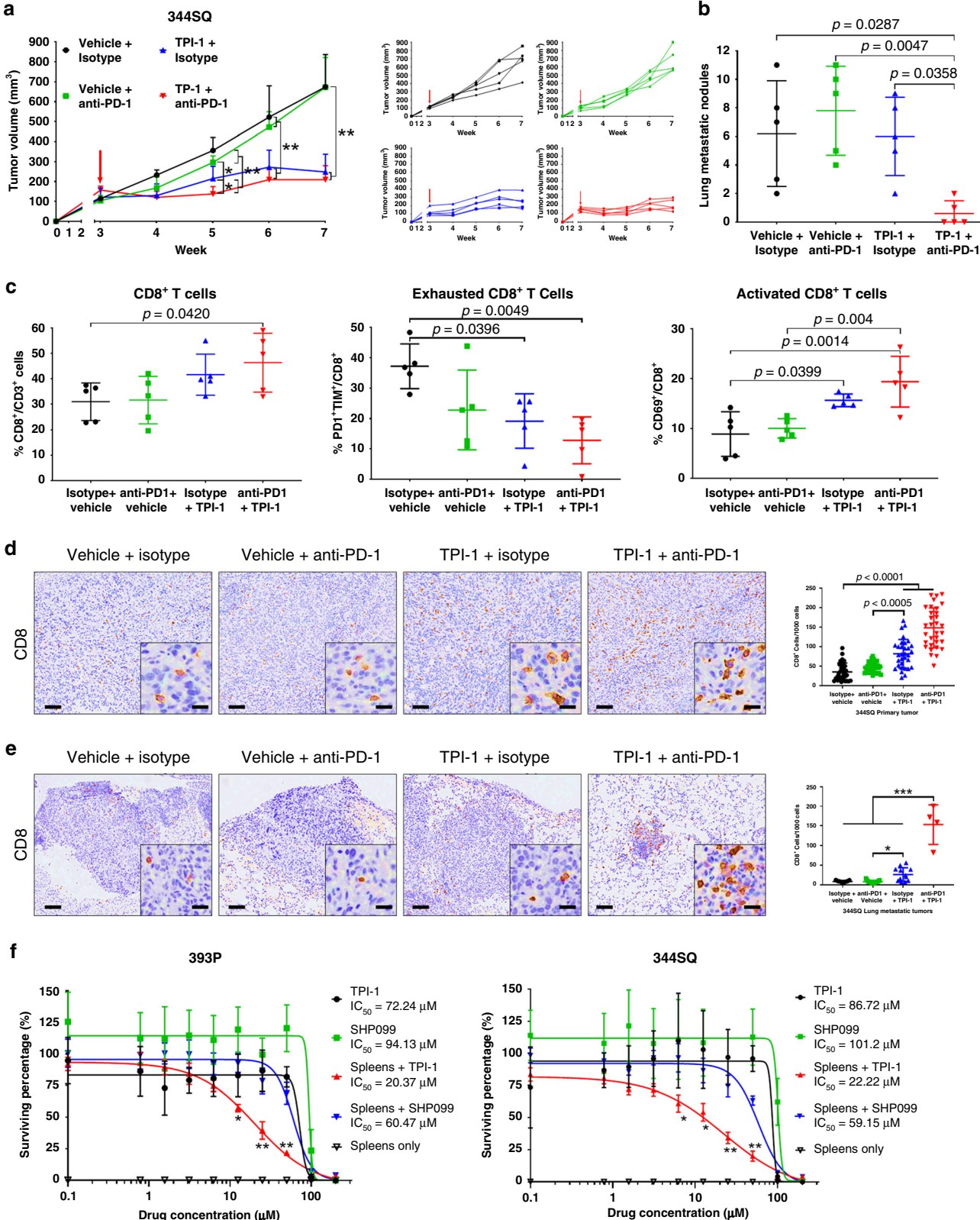

immune cell extravasation to sites of infection including tumors[55,64–66].

The ability to predict response to immune checkpoint blockade based on intratumoral collagen deposition may improve the selection of lung cancer patients likely to benefit from therapy. Moreover, targeting collagen maturation and downstream signaling pathways show the potential for combinatorial strategies with immunotherapies. Lastly, our pre-clinical and clinical findings emphasize the need to develop specific inhibitors of LAIR1 to synergize with PD-1/PD-L1 blockade therapies and emphasize the need to have greater accessibility to lung cancer patient datasets with pre- and post-treatment samples. Owing to the fibrotic nature of many solid cancers, our findings have broader applicability to benefit patients with various different cancer types.

**Fig. 6 Combination of PD-1 blockade with SHP-1 inhibition reduces lung tumor growth and metastasis. a** Left: Tumor volume measurements at indicated time points for 344SQ subcutaneous tumors treated weekly with anti-PD-1 (200 μg/mouse) monotherapy, daily with TPI-1 (3 mg/kg/mouse) monotherapy, or both therapies in combination. Treatment start time denoted by a red arrow. Right: Tumor volume measurements for individual mice in each treatment group from the aforementioned experiment; $n = 5$ mice per treatment group. **b** Quantification of lung metastatic surface nodules in indicated treatment groups at the endpoint of the experiment in (**a**). **c** FACS quantification for percentage of total CD8+ (left), PD-1+TIM-3+ exhausted CD8+ (middle), and CD69+ activated (right) CD8+ TILs in indicated 344SQ tumor cell suspensions from the experiment in (**a**). **d** Representative CD8 IHC stains with quantification of 344SQ primary tumors from the experiment in (**a**); $n = 5$ tumors per group, 32 total fields quantified across all samples for each treatment group. Scale bars, 100 μm. Inset scale bars, 20 μm. **e** Representative CD8 IHC stains with quantification of metastatic lung tumors from the experiment in (**a**); $n = 5$ lungs for isotype + Vehicle (15 total tumor fields), anti-PD1+Vehicle (17 total tumor fields), and isotype + TPI-1 (14 total tumor fields); $n = 2$ lung for anti-PD-1 + TPI-1 with 2 tumor fields per sample analyzed. Scale bars, 100 μm. Inset scale bars, 20 μm. **f** In vitro cell survival response after 72 h of increasing concentrations of SHP-1 (TPI-1) or SHP-2 (SHP099) inhibitor treatment in 393P (left) and 344SQ (right) KP cells alone or in co-culture with splenocytes. Cells were quantified using WST-1 reagent and normalized to DMSO control; $n = 8$ replicates per concentration per group. Indicated statistical significance relative to all treatment groups. All data presented as mean $+/-$ SD. Statistics calculated using a one-way ANOVA post-hoc Tukey test. *$P < 0.05$; **$P < 0.01$.

## Methods

**Cell culture**. Lung cancer cell lines were cultured in RPMI 1640 (Gibco, Thermo Fisher Scientific) supplemented with 10% fetal bovine serum (FBS, Gibco). The 393 P and 344SQ $Kras^{LA1-G12D};p53^{R172H}$ (KP) murine lung cancer cell lines were generated by dissociation of KP lung tumor nodules to single-cell suspensions and seeded on tissue culture plates[35]. The Lewis lung carcinoma LLC-JSP cell line was obtained from ATCC and maintained in our laboratory from past studies[8,12]. HEK-293 cells were obtained from ATCC and cultured in DMEM (Gibco) supplemented with 10% FBS. All cells were cultured at 37 °C in a humidified incubator at 5% $CO_2$ and verified on a monthly basis to be mycoplasma negative using LookOut Mycoplasma PCR Detection Kit (Sigma-Aldrich).

**Plasmids, transfections, and lentiviral generation and transduction**. LOXL2 shRNA constructs were cloned into the pLKO.1-puro vector using primer sequences listed the methods from our prior publication[15], with a scramble sequence as the non-targeting control. LAIR2 open reading frame (ORF) was cloned into the pLenti-puro vector using primers listed in the Supplementary Methods. Stable cell lines were generated using lentiviral transduction, which was first generated by co-transfecting packaging vector psPAX2, envelope vector pMD2.G, and the pLenti-puro expression vector into HEK-293 cells using Lipofectamine LTX. Transfection medium was removed and HEK-293 cells were cultured in RPMI 1640 + 10% FBS for 48 h. Viruses were then syringe-filtered through a 0.45-μm-nylon filter and Polybrene (Santa Cruz) was added to a final concentration of 8 μg/mL. Medium containing lentiviruses was then added to cells, left to allow infection of the cells for 48 h, and replaced with fresh medium for further experiments.

**Mice**. All animal experiments were reviewed and approved by the Institutional Animal Care and Use Committee (IACUC) at The University of Texas MD Anderson Cancer Center. Mice were housed in ventilated cage enclosures in an environment maintained at 50% humidity with ambient temperatures range between 66 °F and 78 °F and 12-h-light/dark cycles. Male and female $Kras^{LSL-G12D};p53^{fl/fl}$ adenovirus-Cre inducible mouse models (129/sv background) of LUAD were infected by intratracheal intubation at 3 months of age. Experiments with adeno-Cre-induced $Kras^{G12D};p53^{-/-}$ mutant mice were performed ~3 months post-infection when tumors properly formed and could be visualized by micro-CT imaging. For experiments with syngeneic tumor xenograft assays, cells were subcutaneously injected in the right flanks of male immunocompetent 129/Sv or BL/6 mice at 3 months of age. For drug treatment experiments, male and female mice were randomly and evenly assigned to designated treatment groups. Mice that required administration of adenovirus were housed specifically in suites designated for biohazard handling as approved under the IACUC protocol. Two weeks post-infection, mice were returned to the regular housing suite. All mice were immunocompetent and assessed for health daily by the Department of Veterinary Medicine and Surgery. All mice were genotyped to determine the mutational status by tail snips 2 weeks after birth.

**In vivo drug response and tumor growth assays**. KP mouse lung cancer cell lines were transplanted into syngeneic wild-type 192/Sv male mice by subcutaneous injection into the right flanks at 3 months of age. LLC-JSP Lewis lung carcinoma cell lines were transplanted in an identical manner into syngeneic wild-type BL/6 male mice. Tumors formed until volumes were ~150–200 mm³ as measured by digital calipers before treatments were administered. Lung tumors from conditional $Kras^{G12D};p53^{-/-}$ mouse models of LUAD were induced in mice by intratracheal delivery of adeno-Cre virus at 3 months of age. Tumors were visualized and analyzed by micro-CT imaging. Mice were treated weekly with 200 μg of PD-1, PD-L1, or CD8 blocking/depletion antibodies (Bio X Cell) by intraperitoneal injection. Ellagic acid was generously provided by Dr. Hal Chapman and administered to mice through ad libitum feeding of chow composed of 2% wt/wt raspberry extract rich in ellagic acid[31]. TPI-1 (Cayman Chemical) was administered daily by oral

gavage at a dosage of 3 mg/kg mouse weight. Control mice received solvent at a volume equal to the drug dosage at the indicated drug concentrations or 200 μg of isotype control (Bio X Cell). Mouse weights were measured weekly to adjust total dosage and assess the effects of drug combinations on mouse health. After euthanasia by $CO_2$ exposure at 3 L/min, syngeneic primary tumors and/or mouse lungs were formalin-fixed, paraffin-embedded, and sectioned for histological analysis. For animal studies, to test the hypothesis that effective combinatorial treatment would reduce primary tumor growth by at least 30% with a variance of 10%, the minimal sample size for 90% power with statistical significance accepted at $\alpha = 0.05$ was four mice per experimental group.

**Reverse-phase protein array (RPPA) preparation**. Tumor tissues were immersed in lysis buffer (1% Triton X-100, 50 mM HEPES [pH 7.4], 150 mM NaCl, 1.5 mM $MgCl_2$, 1 mM EGTA, 100 mM NaF, 10 mM NaPPi, 10% glycerol, 1 mM phenylmethylsulfonyl fluoride, 1 mM $Na_3VO_4$, and protease and phospho-protease inhibitors from Roche) and mechanically homogenized with a motorized tissue homogenizer (Fisher Scientific). Lysates were incubated on ice for 20 min, centrifuged at $18,000 \times g$ for 10 min, and collected for supernatant. Protein concentration was measured using the Pierce BCA Protein Assay Kit (Thermo Fisher Scientific), and protein samples were prepared to a final concentration of 1 μg/μl after mixing with 4× SDS sample buffer (40% glycerol, 8% SDS, 0.25 M Tris-HCl pH 6.8, 10% 2-mercaptoethanol) to produce a 1× SDS sample buffer solution. Protein samples were boiled at 100 °C for 5 min and stored at −80 °C for RPPA processing. RPPA was performed at the MD Anderson Cancer Center core facility and details of the process and antibodies are described here: https://www.mdanderson.org/research/research-resources/core-facilities/functional-proteomics-rppa-core.html.

**Western blotting and qPCR**. Protein lysates were obtained from RPPA samples, boiled in 5× SDS sample buffer with bromophenol blue for 10 min, placed on ice for 10 min, separated by SDS-PAGE, transferred to nitrocellulose membranes, and probed with collagen I antibody (Abcam, ab34710). Total RNA was isolated from cells by TRIzol (Thermo) according to manufacturer protocol and cDNA was generated using qSCRIPT reagents (Quanta). QPCR assays were performed using SYBR Green PCR Master Mix (Thermo) along with primers listed in the Supplementary Methods and normalized to the L32 gene.

**Collagen crosslink analysis**. For collagen crosslink analysis[31], 344SQ tumors with LOXL2 inhibition or knockdown were pulverized in liquid nitrogen, washed with cold PBS and water, lyophilized, and weighed. Samples were reduced with $NaB_3H_4$ and hydrolyzed 6 N HCl. Hydrolysates were analyzed for amino acids and cross-links using liquid chromatography-tandem mass spectrometry (LC-MS/MS). DHLNL, HLNL, and d-Pyr represent unreduced and reduced collagen crosslink forms. Crosslinks were quantified as moles per mole collagen based on the value of 300 residues of hydroxyproline per collagen molecule.

**Flow cytometry**. Subcutaneous tumors were mechanically dissociated followed by enzymatic digestion in RPMI 1640 containing 2 U/mL dispase II (Roche) and 0.5% w/v collagenase type I (Thermo Fisher) at 37 °C for 45 min and tumor cell suspensions were filtered through a 70 μm Falcon Nylon cell strainer (Fisher Scientific). Spleens were mechanically dissociated and filtered through a 40 μm Falcon Nylon cell strainer and rinsed with RPMI 1640. All cell suspensions were centrifuged at $800 \times g$ for 5 min and resuspended in RBC lysis buffer (Biolegend) for 2 min at room temperature. RBC lysis was halted by the addition of RMPI 1640 + 10% FBS and cells were centrifuged and resuspended in FACS buffer (PBS + 2% FBS + 1 mM EDTA). Splenocytes were used for further in vitro co-culture assays. Tumor cell suspensions were counted and $2 \times 10^6$ tumor cells per sample were probed with the indicated fluorescently labeled antibodies listed in the Supplementary Methods. Intracellular stains were performed by fixing cells

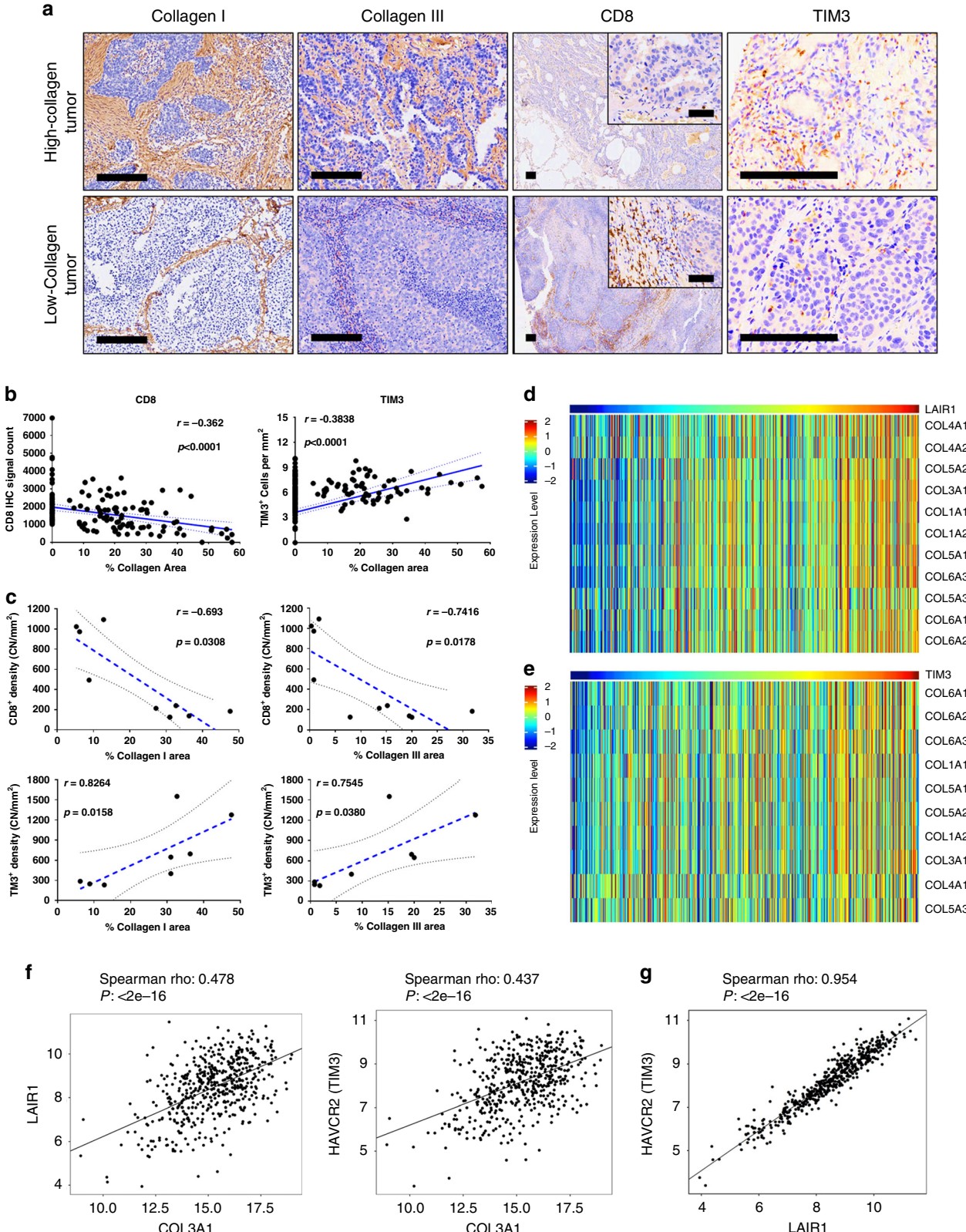

in 1% paraformaldehyde and permeabilizing cells in permeabilization buffer (BD Biosciences) for 30 min at room temperature. Cells were then washed and stained with indicated fluorescently labeled antibodies listed in the Supplementary Methods. FlowJo 10.6.1 was used to analyze FACS data.

**In vitro co-culture and 3D matrigel/collagen culture assays.** Cancer cell lines were seeded in 24-well tissue culture plates and isolated splenocytes were added to

the cells at a 5:1 splenocyte to cancer cell ratio with 1 μg/mL of anti-CD3 and anti-CD28 antibody (Biolegend). Splenocytes were co-cultured with cancer cells for 96 h with indicated treatments and collected from culture media, centrifuged, fixed with 1% paraformaldehyde for 30 minutes at room temperature, centrifuged, and resuspended in FACS buffer containing indicated antibodies for FACS analysis. For 3D Matrigel/collagen culture experiments, 200 μl of Matrigel (Corning) or a 1:1 v/v Matrigel/collagen type I (Corning) mix at 1.5 mg/ml final collagen concentration were coated in 24-well plates. Splenocytes were seeded on the matrices and cultured

**Fig. 7 High-collagen expression correlates with decreased CD8+ T cells, increased LAIR1 and TIM-3 exhaustion markers in human lung cancer datasets. a** Representative images of human lung cancer tissue sections (n = 451 for TMA and 8–10 whole tissue sections) stained by IHC for collagen I, collagen III, CD8, and TIM3. Scale bars, 200 µm. Inset scale bars, 50 µm. **b** Cluster plot analysis of Spearman's rank correlation between total percent collagen I & III area versus CD8 or TIM-3 IHC signal count in human lung cancer tissue microarrays (n = 451 tissue samples). **c** Cluster plot analysis of Spearman's rank correlation between percent collagen I or collagen III area versus CD8 or TIM3 IHC signal count in human lung cancer whole tissue sections (n = 8–10 tissue samples). **d** Heatmap showing the association of mRNA expression in the TCGA LUSC dataset between LAIR1 and collagen genes that are statistically significant (P < 0.05) by Spearman's rank correlation. **e** Heatmap showing the association of mRNA expression in TCGA LUSC dataset between HAVCR2 (TIM-3) and collagen genes that are statistically significant (P < 0.05) by Spearman's rank correlation. **f** Representative cluster plot analysis of Spearman's rank correlation between COL3A1 versus LAIR1 or HAVCR2 (TIM3) mRNA expression in the TCGA LUSC dataset. **g** Cluster plot analysis of Spearman's rank correlation between LAIR1 versus HAVCR2 (TIM3) mRNA expression in the TCGA LUSC dataset.

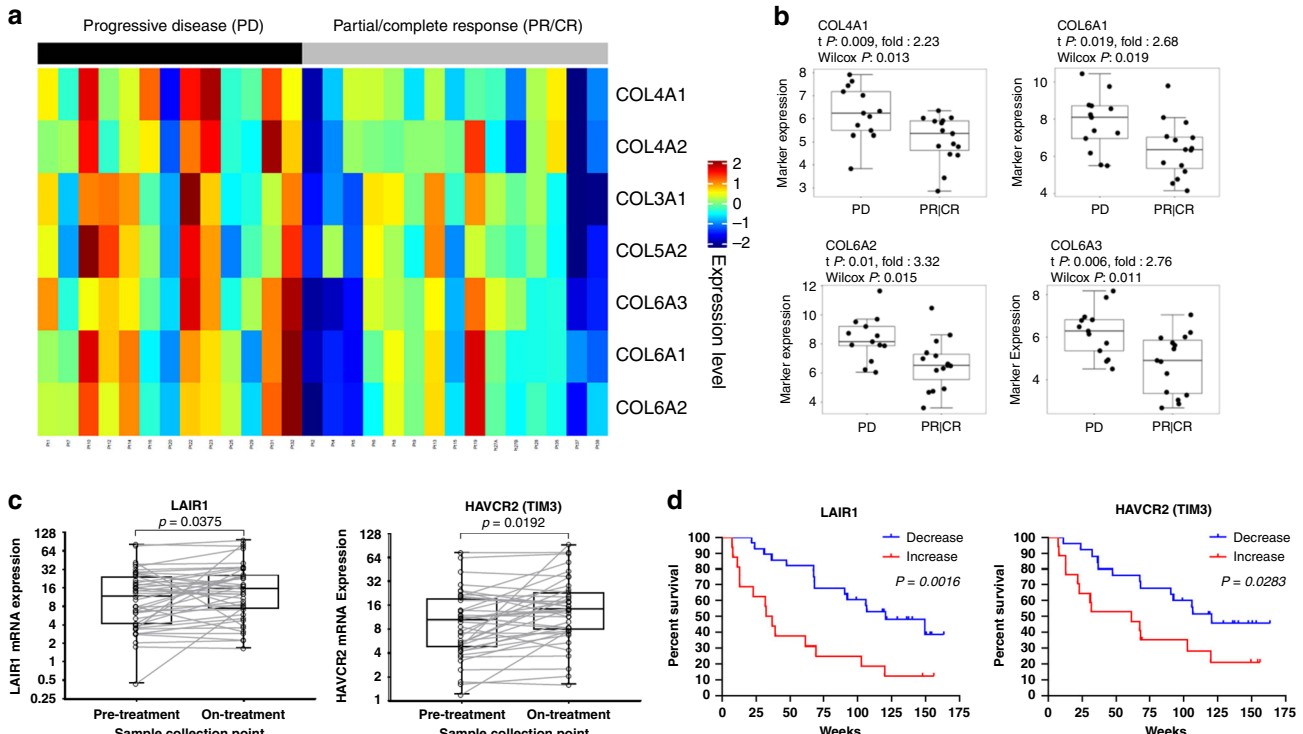

**Fig. 8 Collagen, LAIR1, and TIM3 predict response and overall survival to anti-PD-1/PD-L1 therapies. a** Heatmap showing statistically significant (P < 0.05) association between collagen mRNA expression in pre-treatment tumor biopsies and disease progression (progressive disease vs. partial/complete response) for melanoma patients receiving anti-PD-1 therapy (nivolumab). **b** Representative dot plots comparing pre-treatment collagen mRNA levels in melanoma patients that exhibited progressive disease (PD) versus partial or complete response (PR/CR) to anti-PD-1. PD sample size of n = 13 independent patient samples and PR/CR sample size of n = 15 independent patient samples. Boxplots shown as the median ± 1 quartile, with whiskers extending to the most extreme data point within 1.5 interquartile range from the box boundaries. Statistics calculated using two-sided Wilcoxon matched pair rank test with significance at P < 0.05. **c** Overall changes in LAIR1 and HAVCR2 (TIM3) mRNA expression levels in melanoma tumor biopsy samples between pre- and on-treatment with anti-PD-1; n = 43 independent patient samples for pre- and on-treatment groups. Boxplots shown as the median ± 1 quartile, with whiskers extending to the most extreme data point within 1.5 interquartile range from the box boundaries. Statistics calculated using two-sided Wilcoxon matched pair rank test with significance at P < 0.05. **d** Kaplan–Meier curves predicting survival of melanoma patients receiving anti-PD-1 therapy based on net changes in LAIR1 and HAVCR2 (TIM3) mRNA levels. Statistics calculated using a two-sided log-rank Cox test with significance at P < 0.05.

for 96 h with indicated treatments. Splenocytes and matrices were collected together, fixed, washed, and probed with indicated antibodies for FACS analysis as described. Prior to flow cytometry, labeled splenocytes were filtered into a Falcon 5 mL round-bottom flow tubes with a cell-strainer cap (Corning) to remove residual matrices. For drug response co-culture assays, splenocytes were co-cultured with cancer cells for 72 h, media containing splenocytes were removed, and WST-1 (Roche) colorimetric cell proliferation reagent was added to each well and incubated at 37 °C for 2 h. For splenocytes survival assays, WST-1 was directly added to splenocytes suspended in cell culture media. Color intensity was measured at 450 nm with 690 nm reading subtracted for background. Percent surviving fraction of cells was normalized against untreated cells.

**Histology and SHG microscopy.** Paraffin-embedded tissue sections were rehydrated, stained with hematoxylin and eosin (H&E), IHC, or Masson's trichrome kit (Sigma-Aldrich) following manufacturer protocol, dehydrated, and mounted for

further analysis. For IHC, antigen retrieval was performed after tissue rehydration using citrate buffer, pH 6.0 (Dako Agilent Technologies, Santa Clara, CA) and boiled for 25 min. Endogenous peroxidases were blocked with 3% $H_2O_2$ in TBS and slides were further blocked with 5% goat serum in TBS + 0.1% Tween-20 (TBST). Tissues were probed with LOXL2 or CD8 antibody (Supplementary Methods) at a 1:100 dilution in goat serum overnight at 4 °C. Sections were washed with TBST and incubated with streptavidin-conjugated secondary antibodies targeting rabbit IgG diluted in goat serum for 1 hr at room temperature. Slides were washed again and incubated with biotinylated HRP in goat serum for 30 min at room temperature. After washing, the signal was attained by developing with DAB reagent (Dako) for 5 min at room temperature, immersed in ddH₂O to stop the reaction, and counterstained with Harris Hematoxylin (Thermo Fisher Scientific). For SHG microscopy, tissues stained by H&E were visualized using a Zeiss LSM 7 MP Multiphoton Microscope at an excitation wavelength of 800 nm, and collagen fiber signals were detected at 380–430 nm using bandpass filters. Blue collagen trichrome

signal was isolated and analyzed by ImageJ v1.53, using a color threshold hue between the values of 150 and 190 and the pixel area of collagen was calculated with the ImageJ plugin. The total pixel area of SHG signal was isolated and calculated using the ImageJ plugin. Percent area of collagen was calculated as a ratio of total collagen pixel area to the total pixel area of the tumor tissue image. CD8 IHC quantification was performed by capturing negative images of tissue sections and isolating CD8 cells using a color threshold hue between the values of 120 to 255 with the ImageJ plugin. The ratio of CD8 cells to every 1000 tumor cells was calculated and presented.

**Human lung cancer datasets and lung cancer tissue analyses**. Level 3 mRNA gene expression data from TCGA LUAD and LUSC datasets were analyzed as previously described[15,67,68]. For dataset analysis of human melanoma patients treated with PD-1 blockade, FPKM RNA sequence data previously published from Hugo et al.[47] and Riaz et al.[48] were downloaded from GEO (GSE78220 and GSE91061). Patient samples with 0 sequencing reads were excluded in the analysis. Survival analysis was performed using the Kaplan–Meier method and the log-rank test was used to determine statistical significance. A comparison of gene expression values between response groups was performed using the Wilcoxon matched-pairs signed-rank test.

The second set of samples were obtained from the Profiling of Resistance patterns and Oncogenic Signaling Pathways in Evaluation of Cancers of the Thorax trial. After patient consent and MDACC Institutional Review Board approval of the protocol, retrospectively identified samples from surgical resections from patients who had not received therapy were collected and built into a tissue TMA (1 mm triplicate cores representing each individual tumor)[45,69]. Whole tumor sections from the same set were also utilized. We performed IHC using an automated stainer. Four-micrometer formalin-fixed and paraffin-embedded tissue sections were stained in a Leica Bond Max automated stainer (Leica Biosystems Nussloch GmbH). The tissue sections were deparaffinized and rehydrated following the Leica Bond protocol. Antigen retrieval was performed for 20 min with Bond Solution #2 (Leica Biosystems, equivalent EDTA, pH 9.0) or Bond Solution #1 (Leica Biosystems, equivalent Citrate Buffer, pH6). Primary antibodies were incubated for 15 min at room temperature and detected using the Bond Polymer Refine Detection kit (Leica Biosystems) with DAB as chromogen. The slides were counterstained with hematoxylin, dehydrated, and cover-slipped. Antibody clones and their vendor information, as well as dilution and antigen retrieval conditions, are summarized in the Supplementary Methods. Protein levels for collagen type I and type III were assessed in the entire whole tumor area. Protein levels for TIM-3 and CD8 markers were evaluated in five tumor regions encompassing the tumor inner and peripheral regions and TMA slides were similarly analyzed using the HALO (Indica Labs) software[8,15,45,46,69]. Spearman's Rank Correlation was used to correlate indicated gene expression levels and quantified IHC signal.

**LAIR2 ELISA**. Human LAIR2 ELISA kit (Sigma-Aldrich) was performed according to manufacturer protocol. In brief, conditioned media from cells overexpressing LAIR2 or vector control were added to a 96-well plate pre-coated with LAIR2 capturing antibodies, probed with a biotinylated HRP-linked LAIR2 detection antibody, developed using a chromogenic substrate, and measured at 450 nm with 690 nm reading subtracted for background. Concentrations were calculated based on a LAIR2 standard curve.

**Statistics**. For analysis of RPPA data, a linear mixed model was applied to compare protein expression on a protein-by-protein basis between epithelial and mesenchymal groups; the model includes cell line effects as a random effect factor. The resulting p-values were modeled by a Beta-Uniform Model. To identify differentially regulated mRNA and protein markers between treatment groups, we used a false discovery rate (FDR) of 0.05 as the cutoff. The Pearson correlation was used for distance matrix calculation, and Ward method was applied as a linkage rule for the hierarchical clustering. Spearman's Rank Correlation was used to correlate indicated IHC values or gene expression levels between markers in human patient samples. Statistical significance for all other experiments was assessed using either one-way ANOVA post hoc Tukey test or two-tailed student's t-test, and statistical significance was accepted as having a p-value < 0.05. All experiments were performed in technical triplicates unless otherwise stated. In vivo studies were performed with one replication as sufficient mice samples were obtained based on power analyses to optimize the cost of reagents including mice numbers, animal housing, and drugs. In vitro studies were performed in two or more experimental replicates. Biological replicates are stated in text denoted by sample size N. Statistical analyses and graphs were done using R 3.5.1 and GraphPad 7.

**Reporting summary**. Further information on experimental design is available in the Nature Research Reporting Summary linked to this paper.

## Data availability

Human lung cancer patient data are available from the TCGA repository from https://portal.gdc.cancer.gov/. Treated human melanoma patient data are available in the GEO repository (GSE78220 and GSE91061) from https://www.ncbi.nlm.nih.gov/geo/. The source data for all relevant figures and supplementary figures are provided as a Source Data file with this paper. All other data that support the findings of this study are available upon request to the corresponding author. A reporting summary for this article is available as a Supplementary Information file. Source data are provided with this paper.

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

## Acknowledgements

We thank the Gibbons lab for their input on manuscript preparation and experimental ideas. This work was supported by CPRIT RP150405, NIH R37 CA214609, Uniting Against Lung Cancer/Lung Cancer Research Foundation award, Rexanna's Foundation for Fighting Lung Cancer to D.L.G., CPRIT-MIRA RP160652 (to J.M.K. and D.L.G.), and the UT Lung Cancer SPORE NCI P50 CA070907 (D.L.G., J.M.K., M.Y. and J.W.). J.W. is supported by Cancer Center Support Grant (CCSG CA016672), and Mary K. Chapman Foundation. L.A.B. and D.L.G. are R. Lee Clark Fellows of the University of Texas MD Anderson Cancer Center, supported by the Jeanne F. Shelby Scholarship Fund. The work was also supported by the philanthropic contributions to the University of Texas MD Anderson Lung Cancer Moon Shots Program.

## Author contributions

D.H.P. and D.L.G. conceived the project and designed experiments. D.H.P. wrote the manuscript and performed and/or assisted in all experiments and data analyses with the exception of contributions listed here. B.L.R. and L.C. assisted in the treatment of KP GEMMs and FACS analysis. L.C. provided samples and raw data for the tumor RNA profiling. L.D., L.A.B., and J.W. performed bioinformatics analyses on the RPPA and RNA profile dataset as well as bioinformatics analyses on human TCGA lung cancer patient datasets. Y.W. and H.C. generated and provided the ellagic acid mouse feed. M.Y. performed the collagen crosslink analysis. C.B., G.R., L.M.S.S., and E.R.P.C. performed the IHC on the human tissue samples and also performed the pathological scoring. I.I.W. supervised pathology data acquisition and analyses on human tissue samples. J.M.K. provided additional mice and reagents for in vivo tumor growth experiments. D.L.G. supervised and oversaw all aspects of the project and the writing of the manuscript.

## Competing interests

D.L.G. declares advisory board/consulting work for Janssen, AstraZeneca, GlaxoSmithKline, Astellas, Ribon Therapeutics and Sanofi. D.L.G. receives research grant funding from AstraZeneca, Janssen, Ribon Therapeutics, Astellas and Takeda. L.A.B. declares consulting work for AstraZeneca, AbbVie, GenMab, BergenBio, Pharma Mar, SA. L.A.B. receives research grant funding from AbbVie, AstraZeneca, GenMab, Tolero Pharmaceuticals. All other authors declare that they have no conflict of interests.
