## [Peer Review File · Nature Communications]

Reviewers' comments:

Reviewer #1 (Remarks to the Author); expert in ECM and immune crosstalk:

NCOMMS-19-25894

In this study, Peng et al suggest that PDL1-resistant lung tumours increase collagen deposition with decreases CD8+ T cells and increases CD8+T cell exhaustion. Then authors show that reduction in collagen deposition by LOXL2 inhibition leads to increased CD8+ T cells (and less exhaustion), abrogating PDL1 resistance. Mechanistically, the collagen receptor LAIR1 is increased after ITGB2 interaction with collagen, which then induces T-cell exhaustion through phosphatase SHP1 signalling. Then, LAIR1 inhibition through LAIR2 overexpression or SHP1 inhibition abrogates PDL1 resistance.

While the study presents some interesting pre-clinical data with therapeutic potential, there is a number of issues that dampen this enthusiasm and would preclude publication at this stage. First, statistics are a bit muddled and, in many figures, it is not clear the comparison being made. In some cases there is only 1 asterisk on top of one group (for example, Fig.2b –is this vs control?-, Fig.5a-right panel, Fig.5d-right panel). In cases with more than 2 groups to be compared (most panels in the paper), statistical analysis with multiple comparison correction (ANOVA with post hoc tests) should be employed instead of student's t-test.

Another issue is that the links between collagen deposition and the immune infiltrate, and the potential role of the ECM as a barrier for immune infiltration are not well laid out. This is partly due to different methods used to analyse both parameters -IHC for collagen, FACS for immune infiltrate. Analysis accounting for regional differences would be more informative. For example, in Fig.3b, after LOXL2 inhibition combined with anti-PDL1, are those regions devoid of collagen now more infiltrated with CD8+ cells? Are there differences in the inner regions vs the invasive edge of tumours? Same applies to similar experiments like Fig.5a,d-f and others.

Experiments using LOXL2 inhibitor or LOXL2 shRNA throughout the manuscript do not show efficiency of inhibitor/on-target effect, and efficient knockdown, respectively. In particular, effects of LOXL2 inhibitor are much more potent than the LOXL2 shRNA (Fig.3a vs Supp Fig 3a), so this discrepancy should be addressed. Is LOXL2 expression increased in PD-L1-resistant tumours? For in vivo experiments, individual growth curves should also be shown too, were there any regressions with the treatments?

For the analyses on human samples, the correlations shown are quite weak, except the LAIR1-TIM3 and the integrin-LAIR2. Further, the analyses were done in presumably treatment-naïve patients, is this correct? There is no clinical information regarding treatments for these patients (TMA) or for the TCGA samples used. To strengthen the clinical relevance of the findings, similar analyses (collagen, infiltration) using PDL1- and/or PD1-resistant samples should be performed. In addition, Fig.6b shows a large number of tumours with 0% collagen area. This is misleading since these data came from a TMA, therefore, a very small region that may not be representative of the whole tumour -there is not much info on how this TMA was built either. For this type of analysis, as said above, regional analysis would be more informative, in whole sections if possible, or if in a TMA, this should include several representative areas from tumour core and invasive edge for each tumour.

In general, methods should be more detailed. For example, where the RPPA was performed, which antibodies included, etc. For the patient samples, as said above more information should be included regarding treatments, etc.

Specific points

1. Fig.1- there is some discrepancy in the collagen analyses. Fig 1b shows increased collagen VI, but this is not observed in Fig 1c which shows expression of other collagens increased after antiPDL1. Were those not included in the RPPA? As said above, more detail on this experiments are needed. In Supp.Fig1a an immunoblot for collagen I is shown instead. In general, authors should be more careful with wording, since Masson's staining is used indistinctly with %collagen area, while different collagens are analysed (and not discussed) in other parts of the manuscript (collagen I in Supp.Fig1, collagen III in Fig.6a).

2. As said above, LOXL2 knockdown efficiency should be shown before injection and on tumours by

IHC. Same for LOXL2 inhibitor. Is LOXL2 depletion affecting viability of tumour cells?

3. Fig.2b- It is not clear what the statistical comparison is for. Does shLOXL2 decrease lung metastatic nodules vs control 344SQ and is this statistically significant? Text says "...LOXL2 enzymatic inhibition or knockdown in 344SQ tumours reduced collagen deposition and metastatic disease with no effect on primary tumour growth (Fig. 2b..)".

4. Fig.2c-d: are these data from the lung or subcutaneous tumours? Since LOXL2 shRNA does not affect primary tumour growth but it affects metastatic ability, analysis of infiltrate and collagen deposition should be performed in the lung tumours (by IHC for example).

5. Analysis of other immune infiltrates shows that while CD8+ increase after LOXL2 inhibition, number of CD3+ and CD4+ do not change. Shouldn't the increase in CD8+ be mirrored with increase CD3+?

6. Supplementary Fig.2b images should be quantified too.

7. Fig.3a: as said above, LOXL2 inhibitor is more potent than LOXL2 shRNA, this discrepancy should be addressed, starting with showing on-target effects and knockdown efficacy before injection and on tumours.

8. Fig.3c- authors should explain why %CD8+ is 3% in control while in Fig.2c %CD8+ is 23.8% in control, and In Supp. Fig.3c % CD8+ is 40%, being all the same cell line (334SQ). Are these analyses at endpoint?

Similar discrepancy for %PD1+TIM3+: Fig.2d % in control is 70%, Fig.3d % is 50%, and Supp.Fig.3d % is 25%.

9. For Supp.Fig.4 LLC-JSP model, tumour growth curves should be shown. This model is quite interesting, so if all tumours are regressing with treatment, infiltration analysis should be performed earlier.

10. Fig.4c,d and Supp Fig.5a- starting %CD8+ is very low (4%), is this biologically relevant?

11. Fig.4d-e. Why %CD8+ is 4% in Fig.4d-dms0 plastic, while in Fig.4e vec ctrl is 20%? Is this the effect of the conditioned media from cells?

12. Fig.4. Experiments using SHP1 inhibitors (is the drug on target?) should be validated using RNAi. Is viability of T cells and tumour cells affected?

13. Fig.5a,d - these tumours should be stained for collagen and T cells. Why Fig.5 has antiPD1 while the other figs have antiPD-L1?.

14. Fig.5b- these IC50 values are very high, ranging 20-90uM. This could be off-target. Also as said above in point 13, is T cell viability affected?

15. Fig.5d - there is no synergy between antiPD1 and TPI-1 in primary tumour growth, while there is (supposedly) in lung metastasis. As said above, statistics need revision, but if the metastatic setting is the relevant one in this case, the lung metastasis should be analysed for infiltrates and collagen.

16. Fig.5f - here the only significant comparison is the combination, but this does not correlate with an improvement in primary tumour growth as said above for Fig.5d. Please discuss.

17. Fig.6a. In the patient samples, collagen III was stained -instead of Masson's as in previous experiments, could authors elaborate? There is no collagen III in Fig.1 either.

18. Fig.7 is ITGB2 affected by LOXL2 inhibition or PDL1 blockade? ITGB2 should be stained in tumours, in particular analysing its expression on T cells.

Reviewer #2 (Remarks to the Author); expert in lung cancer:

The authors found that collagen promotes resistance to anti-PD-L1 therapy by causing LAIR1-dependent CD8+ T cell exhaustion. The story they demonstrated is very beautiful.

1. Tumor cells have an innate character to induce collagen by progression and anti PD-L1 therapy.
2. LOXL2 (=collagen stabilizer) inhibition by ellagic acid (EA) for tumor cells down-regulates collagen associated with sensitization to anti PD-L1 therapy through decrease in exhausted CD8+ T cell.

3. LAIR1 and SHP1 inhibition for CD8+ T cell decreases their collagen-mediated exhaustion and

sensitizes anti-PD1 therapy.

4. Expressional relationship among collagen, CD8+ cells, TIM3 (exhausted T marker) and LAIR1 was observed as predicted in human tumor samples.

The results are obtained from their mice models and the results might be biased as the authors discussed on page 14 (i.e., difference in collagen property). However, the reviewer thinks that combined anti-PD-L1/PD-1 and anti-SHP1 therapy is worth investigating in the human.

The following points should be considered.

1. Anti-PD-L1 (Figs 1 and 2) or anti-PD1 (Fig 3) therapy was performed in each mouse experiment. The authors should describe the rationale for their strategy.
2. EGFR or fusion positive adenocarcinoma does not respond well to anti-PD-L1/PD-1 therapies. Omics analysis in Figure 6 might give us some indication.
3. Collagen types are different between in Figure 1 and Figure 6. Do the authors think that rationale obtained in the mice models are reproduced in human tumors?
4. Labeling for ellagic acid should be uniformed between Fig 2 and Fig 3.

Reviewer #3 (Remarks to the Author); expert in immunotherapy:

Authors attempt to demonstrate that murine and human lung tumors resistant to PD-1/PD-L1 immune checkpoint blockade (ICB) have increased levels of collagen, which induces exhaustion of CD8 TILs and reduced CD8 infiltration. The authors propose that this is mediated through SHP-1-mediated LAIR1 induction on CD8 when these interact with collagen through CD18. Using genetic and pharmacologic inhibition of collagen maturation (LOXL2) or SHP-1 in vitro and in vivo, authors demonstrate that tumors become sensitive to the anti tumor effects of ICB. The manuscript is concise and well written. However, some aspects require addressing, as follows.

1. All experiments appear to have been done only once. Please clarify.
2. Using t-tests to examine multi-group comparisons is not adequate. Rather 1-way ANOVA should be used (ex: Fig. 3c, 3d, etc); statistical differences in tumor growth curves should be examined by RM ANOVA.
3. Authors claim that CD8 T cells are the effectors in the proposed mechanism. Please show confirming anti tumor data from a CD8 depletion study (or one performed in nu/nu mice).
4. It is not clear if data presented throughout the manuscript on infiltration of CD8 T cells and other immune subsets is a true infiltration, as data was gated on CD45+CD3. To really clarify this, infiltration data must be shown as "number of live CD8+ T cells/mg of tumor"; same applies to other subsets.
5. It appears throughout the manuscript that the proposed mechanism is claimed as valid for lung cancer. It appears to this reviewer that most of the collagen and immune data is from primary tumor, not lung metastasis. Please clarify this apparent discrepancy and if I am correct, do the main mechanism components (collagen, TILs) hold true in the lung metastasis?
6. The claim that exhausted CD8+ T cells play a role in the mechanism based solely on phenotype (PD1+TIM3+) is weak. If correct, then TCR stimulation of CD8 TILs should show the corresponding results on IL-2/TNF α /IFN γ intracellular cytokine levels. Please address.

Reviewers' comments:

Reviewer #1 (Remarks to the Author); expert in ECM and immune crosstalk:

NCOMMS-19-25894

In this study, Peng et al suggest that PDL1-resistant lung tumours increase collagen deposition with decreases CD8+ T cells and increases CD8+T cell exhaustion. Then authors show that reduction in collagen deposition by LOXL2 inhibition leads to increased CD8+ T cells (and less exhaustion), abrogating PDL1 resistance. Mechanistically, the collagen receptor LAIR1 is increased after ITGB2 interaction with collagen, which then induces T-cell exhaustion through phosphatase SHP1 signaling. Then, LAIR1 inhibition through LAIR2 overexpression or SHP1 inhibition abrogates PDL1 resistance.

While the study presents some interesting pre-clinical data with therapeutic potential, there is a number of issues that dampen this enthusiasm and would preclude publication at this stage.

We thank the reviewer for their support and assistance in improving our manuscript for publication and we have addressed each concern the reviewer has made in a point-by-point manner below.

First, statistics are a bit muddled and, in many figures, it is not clear the comparison being made. In some cases there is only 1 asterisk on top of one group (for example, Fig.2b –is this vs control?-, Fig.5a-right panel, Fig.5d-right panel). In cases with more than 2 groups to be compared (most panels in the paper), statistical analysis with multiple comparison correction (ANOVA with post hoc tests) should be employed instead of student's t-test.

We have revised our statistical tests throughout the manuscript to utilize ANOVA with post-hoc Tukey tests for data with multiple comparisons and updated the text and figure legends accordingly.

Another issue is that the links between collagen deposition and the immune infiltrate, and the potential role of the ECM as a barrier for immune infiltration are not well laid out. This is partly due to different methods used to analyse both parameters -IHC for collagen, FACS for immune infiltrate. Analysis accounting for regional differences would be more informative. For example, in Fig.3b, after LOXL2 inhibition combined with anti-PDL1, are those regions devoid of collagen now more infiltrated with CD8+ cells? Are there differences in the inner regions vs the invasive edge of tumours? Same applies to similar experiments like Fig.5a,d-f and others.

The experiments in Figure 5d-f have moved to Figure 6 of the revised manuscript. We appreciate the Reviewer's suggestion and have performed IHC stains and quantification for the CD8 marker from our *in vivo* experiments shown in Figure 3g-h, Supplementary Figure 6b/d, Figure 5d-e, and Figure 6d-e. Histological stains demonstrate that total CD8 T cell infiltrate is fairly uniform throughout the primary and metastatic tumor tissue, with consistent T cell density in the inner regions and invasive edges of the tumors. The treatment responsive groups devoid of collagen exhibit higher numbers of total CD8 T cell infiltration. LAIR2 overexpression and SHP-1 inhibition did not alter the tumor collagen levels (Supplementary Figure 10c and f), but produced higher CD8 T cell infiltration as assessed by IHC (Figure 5d and Figure 6d), supporting the *in vitro* co-culture assay results wherein collagen prevents CD8 T cell differentiation rather than blocking their infiltration. The histologic patterns are consistent with our observed flow cytometry data from whole processed tumor tissues in all of the *in vivo* experiments.

Experiments using LOXL2 inhibitor or LOXL2 shRNA throughout the manuscript do not show efficiency of inhibitor/on-target effect, and efficient knockdown, respectively. In particular, effects of LOXL2 inhibitor are much more potent than the LOXL2 shRNA (Fig.3a vs Supp Fig 3a), so this discrepancy should be addressed. Is LOXL2 expression increased in PD-L1-resistant tumours?

Supplementary Figure 3a has been moved to Supplementary Figure 5a of the revised manuscript. The LOXL2 inhibitor and shRNA constructs have been previously published [1, 2] and extensively characterized by multiple assays to demonstrate specific LOXL2 on-target effect for both the inhibitor and hairpins. Nonetheless, we have shown in Supplementary Figure 2a-b that the LOXL2 shRNA decreases LOXL2 expression by qPCR and IHC of tumor tissues. This results in decreased total collagen deposition, as shown in Supplementary Figure 2c-d. Additionally, by using mass spectrometry techniques to analyze LOX-dependent collagen crosslinking, as we have previously described and published [2-4], we demonstrate that the LOXL2-specific inhibitor and shRNA significantly reduce collagen crosslinks in the 344SQ tumors. This new data has been added to Figure 2c.

An advantage of using the LOXL2 inhibitor over shRNA is that it is a systemic inhibitor which can prevent collagen deposition from cell sources beyond the tumor cells, such as cancer-associated fibroblasts or immune cells. Moreover, our prior work characterizing the ellagic acid LOXL2 inhibitor [2] demonstrated that ellagic acid binding to the LOXL2 catalytic site produces a by-product that acts as a TGF- β receptor inhibitor, which can further inhibit collagen deposition and TGF- β -induced immunosuppressive effects [5, 6]. Comparison of the results from LOXL2 inhibitor vs the shRNA is further elaborated upon in the revised Discussion section.

LOXL2 is moderately increased overall in 344SQ tumors treated with anti-PD-L1 treatment, as shown in Supplementary Fig. 1a. But LOXL2 expression is already high in 344SQ tumors due to ZEB1 regulation as we previously reported [1] and is one of the reasons we utilized this model to study collagen-mediated PD-L1 blockade resistance.

For in vivo experiments, individual growth curves should also be shown too, were there any regressions with the treatments?

As requested by the Reviewer, we have included the individual growth curves for each treatment group for our *in vivo* experiments in the manuscript (including Figure 3a, Figure 5b, Supplementary Figure 5a, Figure 6a, and Supplementary Figure 7c). In general, we observed good consistency between animals in the same treatment groups and we did not observe complete regression of primary tumors.

For the analyses on human samples, the correlations shown are quite weak, except the LAIR1-TIM3 and the integrin-LAIR2. Further, the analyses were done in presumably treatment-naïve patients, is this correct? There is no clinical information regarding treatments for these patients (TMA) or for the TCGA samples used. To strengthen the clinical relevance of the findings, similar analyses (collagen, infiltration) using PDL1- and/or PD1-resistant samples should be performed.

The Reviewer is correct that in the original manuscript the human patient analyses were all from treatment-naïve patient tumor samples (both the NSCLC TMA and TCGA samples). For the revised manuscript we have also performed IHC stains on whole tumor tissue sections in treatment-naïve lung cancer patient samples, which demonstrate a significantly stronger correlation between collagen, CD8, and TIM-3 staining (Figure 7c).

We completely agree with the reviewer that analyzing patient tumors that have undergone PD-1/PD-L1 blockade treatment would greatly strengthen our clinical findings for this project. To address this issue, we analyzed publically available datasets of melanoma patients treated with anti-PD-1 therapy that have analysis of pre- or matched pre-/on-treatment tumor biopsies. These new data have been added to the revised manuscript and demonstrate that increased pre-treatment collagen levels can predict disease progression and response to anti-PD-1 therapy (Figure 8a and b). Additionally, increasing expression of LAIR1 and TIM-3 following PD-1 blockade predicts poorer overall survival (Figure 8c and d). We analyzed melanoma patient samples because publically available datasets of lung cancer patients treated with immunotherapies with associated pre- and post-treatment biopsies for comparison are lacking, emphasizing the need for these types of studies and sample collection in lung cancer. We have included these points in the revised Discussion section of the manuscript.

In addition, Fig.6b shows a large number of tumours with 0% collagen area. This is misleading since these data came from a TMA, therefore, a very small region that may not be representative of the whole tumour -there is not much info on how this TMA was built either. For this type of analysis, as said above, regional analysis would be more informative, in whole sections if possible, or if in a TMA, this should include several representative areas from tumour core and invasive edge for each tumour.

Figure 6 has now been moved to Figure 7 in the revised manuscript. We thank the reviewer for this suggestion and have stained and analyzed matched whole tumor sections for the indicated collagen and immune markers in Figure 7a,b. Staining and scoring of multiple regions of whole tumor sections shows a significantly stronger inverse correlation between overall collagen I&III vs CD8 and a significantly stronger positive correlation between overall collagen I&III vs TIM3 than for the analyses from the smaller TMA specimens. Additional information detailing the patient tumor datasets, including the TMA construction and associated references, have been added to the Methods section for clarification.

In general, methods should be more detailed. For example, where the RPPA was performed, which antibodies included, etc. For the patient samples, as said above more information should be included regarding treatments, etc.

We have followed the Reviewer's suggestion and have updated the Methods section to provide greater detail throughout. Because the RPPA was performed at the core facility at MD Anderson, we have provided the link to the core's website, which has the full list of the 230+ antibodies used in the RPPA analysis. In addition to the Methods section, we have also provided more details in the revised text of the Results to describe the patient samples.

Specific points

1. Fig.1- there is some discrepancy in the collagen analyses. Fig 1b shows increased collagen VI, but this is not observed in Fig 1c which shows expression of other collagens increased after antiPDL1. Were those not included in the RPPA? As said above, more detail on this experiments are needed. In Supp.Fig1a an immunoblot for collagen I is shown instead. In general, authors should be more careful with wording, since Masson's staining is used indistinctly with %collagen area, while different collagens are analysed (and not discussed) in other parts of the manuscript (collagen I in Supp.Fig1, collagen III in Fig.6a).

We appreciate the Reviewer's comment and have clarified in the text that a limitation of the RPPA is that the antibodies require extensive validation before they can be used for this type of profiling. As such, the other collagen isoforms either do not have an antibody targeting the isoform or the antibody quality is poor for this platform. This is the reason we also analyzed our tumor tissues by RNA profiling. A similar issue applies to our western blot analyses. Many collagen isoforms do not have antibodies that are suitable for westerns but have been validated for IHC of human tissues (e.g. Collagen III). Since we observed multiple upregulated collagen isoforms, we used Masson's trichrome staining and SHG microscopy to encompass total collagen deposition and crosslinking. We have updated our text to be more descriptive and explanatory.

2. As said above, LOXL2 knockdown efficiency should be shown before injection and on tumours by IHC. Same for LOXL2 inhibitor. Is LOXL2 depletion affecting viability of tumour cells?

This point has been addressed above and demonstrated in Figure 2c and Supplementary Figure S2a,b. LOXL2 depletion does not affect tumor viability or growth, as we previously published [1], and shown in Supplementary Figure 5a.

3. Fig.2b- It is not clear what the statistical comparison is for. Does shLOXL2 decrease lung metastatic nodules vs control 344SQ and is this statistically significant? Text says "...LOXL2 enzymatic inhibition or knockdown in 344SQ tumours reduced collagen deposition and metastatic disease with no effect on primary tumour growth (Fig. 2b..)".

The comparison is for the 344SQ Vec Ctrl tumors vs. all other groups. We have updated our graphs throughout the manuscript to be clearer and we changed our statistical analyses to use ANOVA post-hoc Tukey tests for multiple comparisons.

4. Fig.2c-d: are these data from the lung or subcutaneous tumours? Since LOXL2 shRNA does not affect primary tumour growth but it affects metastatic ability, analysis of infiltrate and collagen deposition should be performed in the lung tumours (by IHC for example).

Fig. 2c-d have shifted to Fig. 2d-e in the revised manuscript. These data are from the primary subcutaneous tumors. As requested by the Reviewer, we have performed IHC analysis for CD8 infiltrates on the primary tumors and metastatic lung tissues from the *in vivo* experiments (now included as Figure 3g-h and Supplementary Figure S6d-e), which demonstrates that LOXL2 inhibition or knockdown increases CD8 T cell infiltration into the few remaining metastatic lung tumor nodules.

5. Analysis of other immune infiltrates shows that while CD8+ increase after LOXL2 inhibition, number of CD3+ and CD4+ do not change. Shouldn't the increase in CD8+ be mirrored with increase CD3+?

To clarify this point we have included the gating strategy in Supplementary Figure 3 of the revised manuscript. The increase in the percentage of CD8+ T cells is relative to CD3+ T cells. Not all CD3+ T cells are CD8+, since CD3+ T cells can either be CD4+ or undifferentiated CD8- and CD4-. Therefore, even though CD3+ T cells do not change within the tumor, LOXL2 inhibition or knockdown promotes differentiation of CD3+ T cells to CD8+ T cells. This point that collagen impedes CD3+ T cell differentiation to CD8+ T cells is also demonstrated by our *in vitro* co-culture assay data (Fig. 4, Supplementary Fig. 8a and 9a).

6. Supplementary Fig.2b images should be quantified too.

Supplementary Fig. 2b is now Supplementary Fig. 2c and has been quantified and graphed in Supplementary Fig. 2d.

7. Fig.3a: as said above, LOXL2 inhibitor is more potent than LOXL2 shRNA, this discrepancy should be addressed, starting with showing on-target effects and knockdown efficacy before injection and on tumours.

The on-target effect has now been addressed in Figure 2c, Supplementary Fig. 2a,b, and our previous publications [1, 2]. The advantages of using a systemic LOXL2 inhibitor is that it inhibits LOXL2 from non-tumor cell sources, such as cancer-associated fibroblasts, and because the inhibitor has a LOXL2-dependent by-product that is a TGF- β receptor inhibitor, preventing the immunosuppressive effects of TGF- β and tumor cell-independent sources of collagen. We have now addressed this point in the manuscript Discussion.

8. Fig.3c- authors should explain why %CD8+ is 3% in control while in Fig.2c %CD8+ is 23.8% in control, and In Supp. Fig.3c % CD8+ is 40%, being all the same cell line (3345Q). Are these analyses at endpoint?

Similar discrepancy for %PD1+TIM3+: Fig.2d % in control is 70%, Fig.3d % is 50%, and Supp.Fig.3d % is 25%.

We thank the reviewer for pointing out this discrepancy, which arises partially from differences in the analyses performed at experimental endpoints as euthanasia times are dictated by vet staff and IACUC procedures. Additionally, the discrepancy between the percentages in different immune populations is due to the differences in gating strategy between experiments. To address this issue we have reanalyzed our FACS data using a uniform gating scheme (as outlined in Supplementary Fig. 3) for the *in vivo* experiments so that all the analyses and data are now more consistent across the revised manuscript.

9. For Supp.Fig.4 LLC-JSP model, tumour growth curves should be shown. This model is quite interesting, so if all tumours are regressing with treatment, infiltration analysis should be performed earlier.

The data in Supplementary Fig. 4 has now moved to Supplementary Fig. 7 of the revised manuscript. We have added the individual tumor growth curves for the LLC-JSP model in Supplementary Figure 7b,c.

10. Fig.4c,d and Supp Fig.5a– starting %CD8+ is very low (4%), is this biologically relevant?

Supplementary Fig. 5a is now Supplementary Fig. 8a of the revised manuscript. These T cells were analyzed from untreated wild-type mice splenocytes approximately 6 to 8 weeks of age without exposure to foreign antigens, so percentages of CD8+ T cell differentiation are expected to be lower.

11. Fig.4d-e. Why %CD8+ is 4% in Fig.4d-dmsO plastic, while in Fig.4e vec ctrl is 20%? Is this the effect of the conditioned media from cells?

These are now Figure 4c-d in the revised manuscript, as Figure 4e represents tumor data. The Reviewer is correct that the difference is due to conditioned media. The Vec Ctrl conditioned media is from 3445Q cell culture media and exposure of the splenocytes to the conditioned media increases the percentages of CD8 T cells.

12. Fig.4. Experiments using SHP1 inhibitors (is the drug on target?) should be validated using RNAi. Is viability of T cells and tumour cells affected?

The TPI-1 SHP-1 inhibitor has been published and reported to have exponentially higher affinity for SHP-1 compared to SHP-2 and other targets at the concentrations we used *in vitro* and *in vivo* [7]. We agree that RNAi would provide an additional validation. However, since we are studying SHP-1 signaling in immune cell populations both *in vitro* and *in vivo*, transfection or transduction of RNAi (siRNA or shRNA) into immune cells is not feasible to sustain for the times required for our *in vitro* assays and for *in vivo* experiments.

Our FACS analyses for *in vivo* experiments gate for “Live” immune cell populations, so all of the analyzed T cells are live (Supplementary Fig. 3). Additionally, TPI-1 SHP-1 inhibition shows an increasing trend in total and activated CD8+ tumor infiltrating T cells (Figure 6c), which is statistically significant in the combination treatment group. To further address this concern, we have also performed WST-1 cell viability assays with *in vitro* splenocyte co-cultures in the presence or absence of 20 μ M TPI-1 and observed no significant decrease in viable splenocytes (Supplementary Fig. 8b).

13. Fig.5a,d – these tumours should be stained for collagen and T cells. Why Fig.5 has antiPD1 while the other figs have antiPD-L1?

Figure 5a and d is now Figure 5a and Figure 6a of the revised manuscript. We appreciate the Reviewer’s suggestion and tumors from both experiments have been stained by Masson’s trichrome for collagen in Supplementary Fig. 10c and f and show no observable difference in collagen levels in each treatment group. CD8 IHC has also been performed in matching tumor tissue sections, showing increased levels of CD8 tumor infiltration both in the invasive edge and inner regions of the tumors when inhibition of LAIR1 signaling is combined with PD-1 blockade, consistent with our observed FACS analyses.

In addition to our consistent observations that anti-PD-L1 and anti-PD-1 have similar anti-tumor and pro-immune effects in these models [8, 9] (Figure 3a-f), we have now repeated the LOXL2 inhibitor experiment in combination with anti-PD-1 to address the reviewer’s comment. This new data is shown in Figure 3d. LOXL2 inhibition in combination with anti-PD-1 shows similar pattern of tumor reduction compared to LOXL2 inhibitor + anti-PD-L1. We chose anti-PD1 for our downstream experiments as PD-1 blockade has been shown to be more effective clinically for lung cancer patient survival.

14. Fig.5b- these IC50 values are very high, ranging 20-90 μ M. This could be off-target. Also as said above in point 13, is T cell viability affected?

Figure 5b is now Figure 6f of the revised manuscript. The TPI-1 concentration of \sim 20 μ M is well within the range of concentrations that the original study [7] used for *in vitro* splenocyte and cell culture assays, which demonstrated minimal off-target effects and cell viability. As mentioned above, to address this concern, we treated splenocytes with 20 μ M TPI-1 and observed no significant effect on splenocyte or T cell viability (Supplementary Fig. 8b). Additionally, the high TPI-1 IC50 values in tumor cells alone was expected as SHP-1 inhibition has been previously reported to have no significant effect on tumor cell viability, indicating that TPI-1-mediated reduction in tumor growth is immune-dependent.

15. Fig.5d – there is no synergy between antiPD1 and TPI-1 in primary tumour growth, while there is (supposedly) in lung metastasis. As said above, statistics need revision, but if the metastatic setting is the relevant one in this case, the lung metastasis should be analysed for infiltrates and collagen.

Figure 5d is now Figure 6a of the revised manuscript. As requested, the statistics have been modified and the graphs have been updated. Additionally, we have performed CD8 IHC stains on the lung tumor

tissues from the experiment. TPI-1 inhibition has no observable effect on collagen deposition (Supplementary Fig. 10f). Histological analyses of lung tissues consistently revealed that lung tumor nodules were fewer and much smaller in size in the TPI-1+anti-PD1 treatment group. Additionally, IHC stains revealed markedly higher infiltration of CD8 T cells in the metastatic lung tumors in the combination treatment group (Figure 6e). This finding corroborates our earlier studies that the KP primary and metastatic tumors may have variable response to different therapies, either owing to total size of the tumor cell mass or other differences in the site of colonization.

16. Fig.5f – here the only significant comparison is the combination, but this does not correlate with an improvement in primary tumour growth as said above for Fig.5d. Please discuss.

Figure 5d and f are now Figure 6a and c of the revision. We thank the reviewer for pointing out this error in the figure. As outlined above, we have modified our flow gating strategy to be uniform across all the *in vivo* experiments, along with modifying our statistical analyses. As such, we observe a statistically significant decrease in exhausted CD8+ TILs when tumors are treated with TPI-1 as a single-agent or in combination with anti-PD1. Additionally, we also observe a significant increase in activated CD69+ CD8 TILs when tumors are treated with TPI-1, which is consistent with the new IHC stains for CD8 T cells in Figure 6d.

17. Fig.6a. In the patient samples, collagen III was stained -instead of Masson's as in previous experiments, could authors elaborate? There is no collagen III in Fig.1 either.

Figure 6a is now Figure 7a of the revision. Because of the limited numbers of whole tumor sections available, our pathology collaborators used previously validated/published collagen IHC antibodies [1] to minimize loss of specimens required for optimization. We have also added collagen I stains to the patient tumor whole section analyses, which demonstrates similar correlations across multiple collagen isoforms. The collagen III antibody only works for IHC applications, which we have used in our tumor models as previously published [1]. Because we observed multiple collagen isoforms upregulated following resistance to PD-L1 blockade in the animal models, we used trichrome stains to encompass total collagens in those experiments.

18. Fig.7 is ITGB2 affected by LOXL2 inhibition or PDL1 blockade? ITGB2 should be stained in tumours, in particular analysing its expression on T cells.

Figure 7 has now been merged into Figure 4g,h for the revised manuscript. ITGB2 is ubiquitously and specifically expressed in leukocytes. The main focus of our manuscript was to identify potential therapeutic targets to abrogate resistance to anti-PD-1/PD-L1 therapies. From our studies, we validate LOXL2, LAIR1, and SHP-1 as novel candidates to combine with immunotherapy. The purpose of our experiments with ITGB2 was to identify the mechanism of collagen upregulation of LAIR1. Unfortunately, as mentioned in the discussion, ITGB2 is necessary for immune cell extravasation into tissues during normal infection responses and multiple studies have demonstrated that targeting ITGB2 has failed clinically and is detrimental to patients [10]. Though we agree with the reviewer that understanding the regulation of ITGB2 is interesting, that mechanistic study goes beyond the scope of our current manuscript.

Reviewer #2 (Remarks to the Author); expert in lung cancer:

The authors found that collagen promotes resistance to anti-PD-L1 therapy by causing LAIR1-dependent CD8+ T cell exhaustion. The story they demonstrated is very beautiful.

1. Tumor cells have an innate character to induce collagen by progression and anti PD-L1 therapy.
2. LOXL2 (=collagen stabilizer) inhibition by ellagic acid (EA) for tumor cells down-regulates collagen associated with sensitization to anti PD-L1 therapy through decrease in exhausted CD8+ T cell.
3. LAIR1 and SHP1 inhibition for CD8+ T cell decreases their collagen-mediated exhaustion and sensitizes anti-PD1 therapy.
4. Expressional relationship among collagen, CD8+ cells, TIM3 (exhausted T marker) and LAIR1 was observed as predicted in human tumor samples.

The results are obtained from their mice models and the results might be biased as the authors discussed on page 14 (i.e., difference in collagen property). However, the reviewer thinks that combined anti-PD-L1/PD-1 and anti-SHP1 therapy is worth investigating in the human.

We are very thankful and appreciative of the reviewer for their support and assistance in improving our manuscript for publication and we have addressed each concern in a point-by-point manner below:

The following points should be considered.

1. Anti-PD-L1 (Figs 1 and 2) or anti-PD1 (Fig 3) therapy was performed in each mouse experiment. The authors should describe the rationale for their strategy.

For Figure 1, we treated tumors with anti-PD-L1 because we wanted to identify changes in protein expression by RPPA (Figure 1b) in resistant tumors and cross-reference that data with a previously published RNA profile shown in Figure 1c in which we have treated the same tumor model with anti-PD-L1 [8]. We have repeated similar experiments using the LOXL2 inhibitor with anti-PD-1, which shows a similar pattern as anti-PD-L1 blockade in combination with LOXL2 inhibition (Figure 3d). We ultimately selected anti-PD-1 for the LAIR2 and SHP-1 inhibitor combination treatments because PD-1 blockade shows better clinical efficacy and is being used in broader combinations for patients with lung cancer and melanoma.

2. EGFR or fusion positive adenocarcinoma does not respond well to anti-PD-L1/PD-1 therapies. Omics analysis in Figure 6 might give us some indication.

We appreciate this suggestion and have compared expression of the main collagen isoforms between mutant EGFR and KRAS human tumors in TCGA datasets, which shows no significant difference in collagen expression between the tumor groups. This data has been added as Supplementary Figure 11e-f in the revised manuscript. Future studies with mutant EGFR murine models may be required to further elucidate the resistance mechanisms to anti-PD-L1/-PD-1 therapies beyond what has been published.

3. Collagen types are different between in Figure 1 and Figure 6. Do the authors think that rationale obtained in the mice models are reproduced in human tumors?

Figure 6 has been moved to Figure 7 of the revised manuscript. We agree with the reviewer about the discrepancy in collagen isoforms, which are dependent on available and suitable antibodies for the specific analyses. Nonetheless, we have now stained the human patient samples with collagen I and this

data has been added in Figure 7a/c. We have shown that total collagen and collagen I are upregulated in anti-PD-L1-resistant tumors (Figure 1 and Supplementary Fig. 1a). IHC stains for collagen isoforms in human patient tissues are limited by the antibodies which have been optimized for this protocol. We believe that the rationale obtained in our mice models are reproduced in human tumors as now demonstrated in whole tissue section analyses (Figure 7c), TCGA dataset analyses (Figure 7f,g), and anti-PD-1-treated patients (Figure 8).

4. Labeling for ellagic acid should be uniformed between Fig 2 and Fig 3.

We appreciate the Reviewer pointing this out to us. The figures have been corrected to be uniform throughout.

Reviewer #3 (Remarks to the Author); expert in immunotherapy:

Authors attempt to demonstrate that murine and human lung tumors resistant to PD-1/PD-L1 immune checkpoint blockade (ICB) have increased levels of collagen, which induces exhaustion of CD8 TILs and reduced CD8 infiltration. The authors propose that this is mediated through SHP-1-mediated LAIR1 induction on CD8 when these interact with collagen through CD18. Using genetic and pharmacologic inhibition of collagen maturation (LOXL2) or SHP-1 in vitro and in vivo, authors demonstrate that tumors become sensitive to the anti tumor effects of ICB. The manuscript is concise and well written. However, some aspects require addressing, as follows.

We thank the reviewer for their support and assistance in improving the manuscript for publication and we have addressed each concern the reviewer has made in a point-by-point manner below.

1. All experiments appear to have been done only once. Please clarify.

We have previously performed statistical power analyses on our tumor models, described in the Methods section, to obtain an appropriate number of biological replicates necessary to test the hypothesis while minimizing the cost of reagents including mice and drugs. As further described in more detail in the Methods section, our experiments using this model system are performed similar to previous published work by our group. Additionally, most of the experiments have the same control groups, variable groups and treatment arms that are repeated across multiple experiments, such as the anti-PD1/PD-L1 treatment groups or *in vitro* co-cultures with splenocytes in different ECM.

2. Using t-tests to examine multi-group comparisons is not adequate. Rather 1-way ANOVA should be used (ex: Fig. 3c, 3d, etc); statistical differences in tumor growth curves should be examined by RM ANOVA.

We appreciate the Reviewer's comment and we have adjusted our statistical analyses to have 1-way ANOVA with post-hoc Tukey tests for multi-group comparisons and repeated measures ANOVA in tumor growth curves, which shows similar trends in statistical significance across experiments. We have modified and clarified our methods in the text.

3. Authors claim that CD8 T cells are the effectors in the proposed mechanism. Please show confirming anti tumor data from a CD8 depletion study (or one performed in nu/nu mice).

We appreciate this suggestion from the Reviewer and have performed a repeat of our ellagic acid LOXL2 inhibitor + anti-PD-1 treatment assay in mice with anti-CD8 depletion. This new data has been added as

Figure 3d of the revised manuscript. From this experiment, LOXL2 inhibition with anti-PD1 shows similar reduction in tumor growth as compared to LOXL2i + anti-PD-L1. However, when CD8 T cells were depleted, tumor growth rate was similar to the control group, supporting our proposed mechanism that tumor reduction is dependent upon CD8 T cells.

4. It is not clear if data presented throughout the manuscript on infiltration of CD8 T cells and other immune subsets is a true infiltration, as data was gated on CD45+CD3. To really clarify this, infiltration data must be shown as "number of live CD8+ T cells/mg of tumor"; same applies to other subsets.

We have clarified and modified the gating strategy for all *in vivo* experiments as shown in Supplementary Fig. 3. For all tumor FACS analyses, all cell populations were gated for "Live" cells prior to gating for CD45+ immune subsets, thus all CD8+ T cells analyzed are "live." Additionally, our tissue processing for FACS analysis normalizes for similar number of live/viable cells per tumor (2×10^6 viable cells) for each experimental group. As a result of our processing and gating techniques, the CD8 T cell trends remain similar even with re-analysis.

Nonetheless, we completely agree with the reviewer that true CD8 T cell infiltration could be better demonstrated. Thus, we have performed IHC staining and quantification for CD8 markers in primary and lung metastatic tumor tissues from all *in vivo* treatment experiments (now included in Figure 3g-h, Supplementary Figure S6b-c, Figure 5d-e, and Figure 6d-e). Our CD8 IHC stains demonstrate that when tumors are responsive to therapies, there is an increase in total CD8 T cell infiltration into the primary and metastatic tumor tissues throughout the periphery and inner regions of the tumor.

5. It appears throughout the manuscript that the proposed mechanism is claimed as valid for lung cancer. It appears to this reviewer that most of the collagen and immune data is from primary tumor, not lung metastasis. Please clarify this apparent discrepancy and if I am correct, do the main mechanism components (collagen, TILs) hold true in the lung metastasis?

We have clarified throughout the manuscript the tissue source of the immune data. The reviewer is correct that in the original submission the immune data from the *in vivo* experiments was from the primary tumor and not lung metastasis. To address the Reviewer's point, we have performed CD8 IHC analysis on lung metastatic tumor tissues (Figure 3h, Supplementary Figure S6c, Figure 5e, and Figure 6e), which shows a similar pattern to that observed in the primary tumors – tumors that are responsive to treatment have increased CD8 T cell infiltration in the lung metastatic nodules.

6. The claim that exhausted CD8+ T cells play a role in the mechanism based solely on phenotype (PD1+TIM3+) is weak. If correct, then TCR stimulation of CD8 TILs should show the corresponding results on IL-2/TNF α /IFN γ intracellular cytokine levels. Please address.

We appreciate the Reviewer's suggestion and have addressed this issue by performing flow analysis of intracellular IFN γ in CD8+ TILs from the experiment in Figure 3d. When tumors are responsive to combinatorial treatment, we observe a significant increase in intracellular IFN γ in CD8+ TILs (Figure 3f) that corresponds to a decrease in PD1+/TIM3+/CD8+ T cells. We do not observe an increase in IL-2 in the combination group (Supplementary Fig. 6a). We agree with the reviewer that decreased PD1+/TIM3+ T cells is insufficient for decreased tumor growth, which is why knockdown or inhibition of LOXL2 alone does not significantly reduce tumor burden. Thus, for experiments involving LAIR2 overexpression or SHP-1 inhibition, we analyzed additional CD8 T cell subsets and observed an increase in either CD44+

effector CD8 T cells or CD69+ CD8 T cells when anti-PD1 was combined with LAIR2 overexpression or SHP-1 inhibition, respectively. Both subsets of CD8 T cells have been reported to secrete high levels of IFN γ .

We would like to extend our gratitude to all reviewers for their contributions to make this manuscript suitable for publication.

References

1. Peng, D.H., et al., *ZEB1 induces LOXL2-mediated collagen stabilization and deposition in the extracellular matrix to drive lung cancer invasion and metastasis*. *Oncogene*, 2017. **36**(14): p. 1925-1938.
2. Wei, Y., et al., *Fibroblast-specific inhibition of TGF-beta1 signaling attenuates lung and tumor fibrosis*. *J Clin Invest*, 2017. **127**(10): p. 3675-3688.
3. Chen, Y., et al., *Lysyl hydroxylase 2 induces a collagen cross-link switch in tumor stroma*. *J Clin Invest*, 2015. **125**(3): p. 1147-62.
4. Yamauchi, M., et al., *The fibrotic tumor stroma*. *J Clin Invest*, 2018. **128**(1): p. 16-25.
5. Chakravarthy, A., et al., *TGF-beta-associated extracellular matrix genes link cancer-associated fibroblasts to immune evasion and immunotherapy failure*. *Nat Commun*, 2018. **9**(1): p. 4692.
6. Mariathasan, S., et al., *TGFbeta attenuates tumour response to PD-L1 blockade by contributing to exclusion of T cells*. *Nature*, 2018. **554**(7693): p. 544-548.
7. Kundu, S., et al., *Novel SHP-1 inhibitors tyrosine phosphatase inhibitor-1 and analogs with preclinical anti-tumor activities as tolerated oral agents*. *J Immunol*, 2010. **184**(11): p. 6529-36.
8. Chen, L., et al., *CD38-Mediated Immunosuppression as a Mechanism of Tumor Cell Escape from PD-1/PD-L1 Blockade*. *Cancer Discov*, 2018. **8**(9): p. 1156-1175.
9. Chen, L., et al., *Metastasis is regulated via microRNA-200/ZEB1 axis control of tumour cell PD-L1 expression and intratumoral immunosuppression*. *Nat Commun*, 2014. **5**: p. 5241.
10. Dove, A., *CD18 trials disappoint again*. *Nat Biotechnol*, 2000. **18**(8): p. 817-8.

REVIEWERS' COMMENTS:

Reviewer #1 (Remarks to the Author):

Authors have addressed all my concerns.

Reviewer #2 (Remarks to the Author):

The authors have properly revised the manuscript on the issues pointed out by the reviewer.

Reviewer #3 (Remarks to the Author):

Authors have adequately and thoroughly addressed all my concerns. The manuscript is highly improved.

Dear Reviewers:

Reviewer #1 (Remarks to the Author):

Authors have addressed all my concerns.

We thank the reviewer for their support and assistance in improving our manuscript for publication.

Reviewer #2 (Remarks to the Author):

The authors have properly revised the manuscript on the issues pointed out by the reviewer.

We thank the reviewer for their support and assistance in improving our manuscript for publication.

Reviewer #3 (Remarks to the Author):

Authors have adequately and thoroughly addressed all my concerns. The manuscript is highly improved.

We thank the reviewer for their support and assistance in improving our manuscript for publication.

We would like to extend our gratitude to all reviewers for their input and insightful comments to make this manuscript suitable for publication.